# GeoSDF: Plane Geometry Diagram Synthesis via Signed Distance Field

## Abstract

Plane Geometry Diagram Synthesis has been a crucial task in computer graphics, with applications ranging from educational tools to AI-driven mathematical reasoning. Traditionally, we rely on manual tools (e.g., Matplotlib and GeoGebra) to generate precise diagrams, but this usually requires huge, complicated calculations. Recently, researchers start to work on model-based methods (e.g., Stable Diffusion and GPT5) to automatically generate diagrams, saving operational cost but usually suffering from limited realism and insufficient accuracy. In this paper, we propose a novel framework, GeoSDF, to automatically generate diagrams efficiently and accurately with Signed Distance Field (SDF). Specifically, we first represent geometric elements (e.g., points, segments, and circles) in the SDF, then construct a series of constraint functions to represent geometric relationships. Next, we optimize those constructed constraint functions to get an optimized field of both elements and constraints. Finally, by rendering the optimized field, we can obtain the synthesized diagram. In our GeoSDF, we define a symbolic language to represent geometric elements and constraints, and our synthesized geometry diagrams can be self-verified in the SDF, ensuring both mathematical accuracy and visual plausibility. In experiments, through both qualitative and quantitative analysis, GeoSDF synthesized both normal high-school level and IMO-level geometry diagrams. We achieve 88.67% synthesis accuracy by human evaluation in the IMO problem set. Furthermore, we obtain a very high accuracy of solving geometry problems (over 95% while the current SOTA accuracy is around 75%) by leveraging our self-verification property. All of these demonstrate the advantage of GeoSDF, paving the way for more sophisticated, accurate, and flexible generation of geometric diagrams for a wide array of applications. The accompanying code, datasets, and all synthesized outputs are being released to benefit the research community.

## 1 Introduction

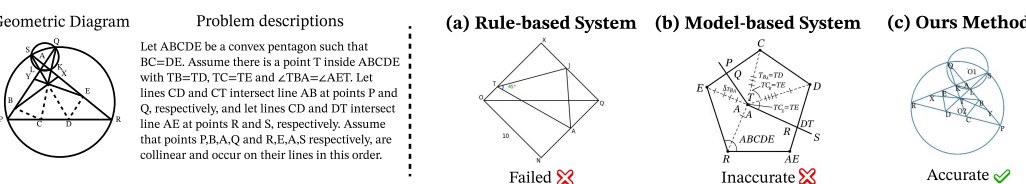

Figure 1: Comparison of geometric diagram synthesis methods for IMO 2022, Problem 4. (a) Rule-based systems, such as R-COT, rely on predefined templates and fail to produce diagrams when faced with non-predefined geometric constraints. (b) General model-based systems, including Gemini, DALL-E 3, and GPT-4o, often lack the precision for accurate mathematical synthesis, resulting in geometrically inconsistent figures. (c) In contrast, our proposed method accurately constructs the diagram by interpreting the fundamental geometric elements and constraints directly from the problem description.

The study of geometry problems is fundamental across numerous disciplines, serving as a corner-stone for advancements in computer graphics Ye et al. (2020), computational geometry Dorst &

Mann (2002), and even theoretical mathematics Trinh et al. (2024). The emergence of large language models (LLMs) has introduced new approaches for addressing Plane Geometry Problems (PGPs) Shi et al. (2024) by understanding both textual problem descriptions and their corresponding geometric diagrams Gao et al. (2025). Despite significant efforts to enhance the reasoning capabilities of LLMs, the performance of models remains limited by the scarcity of high-quality datasets containing annotated plane geometry diagrams Zhang et al. (2023b).

There are several techniques designed to improve the construction of geometric diagrams. A straightforward approach is manual construction, such as Matplotlib Hunter (2007) and GeoGebra Team (2024). However, this process is often cumbersome and time-consuming, as it requires explicit specification of element positions and offers limited flexibility for handling complex configurations. To accelerate diagram generation, rule-based approaches have been proposed Kazemi et al. (2024), which employ programmatic techniques to automate construction. Nonetheless, these methods rely heavily on predefined geometric primitives and fixed assembly rules, which pose limitations when generating diagrams with complex or irregular configurations. The remarkable capabilities of model-based methods like Stable Diffusion Rombach et al. (2021), GPT5, and DALL·E 3 OpenAI (2024) in producing realistic natural images offer an important insight: it raises the question of whether similar approaches can be leveraged to generate geometric diagrams. Unfortunately, as illustrated in Figure 1, these methods struggle with the deterministic rules and mathematical validity (*e.g.,* exact coordinates, relations), resulting in inaccurate diagrams Zhang et al. (2023a;b).

In this paper, we propose a novel framework to automatically synthesize precise **Geo**metry diagrams by leveraging **S**igned **D**istance **F**ield (SDF), termed as **GeoSDF**. In the first step, problem descriptions are parsed into a structured symbolic representation consisting of **elements** (e.g., points, lines, circles) as well as the relationship among them, which are termed as **constraints**. We then represent each geometric element as a signed distance field (SDF), a continuous and differentiable function that compactly encodes its shape and position. Geometric constraints are expressed as differentiable functions over these fields, allowing us to formulate a global loss that enforces all constraints simultaneously. To prevent degenerate configurations such as overlapping elements, we additionally introduce a crowd regularization term that encourages well-spaced solutions. The resulting differentiable objective can be optimized using standard gradient-based methods. Finally, the optimized geometric elements are rendered into clear, human-readable diagrams, providing precise diagrams that faithfully reflect the mathematical relationships.

Previous works Ye et al. (2020) adopted a hard learning curve of domain-specific languages; in contrast, we leverage Large Language Models (LLMs) to translate natural language directly into a symbolic format. Based on simple use, our framework has the state-of-the-art performance on downstream tasks. We demonstrate the effectiveness on a range of challenging datasets, from the high-school level to an IMO-level benchmark. Our method achieves a problem-solving accuracy of 95.9%, a high-confidence result that confirms the reliability of our tool and surpasses the previous best by over 20 percentage points on the GeoQA dataset. And, our method enables the successful synthesis of diagrams for IMO-level problems where prior methods fall short. Moreover, our optimization process operates in batches, enabling the highly efficient, parallel synthesis. The contributions of this paper are summarized as follows:

- We introduce GeoSDF, a novel and accurate framework for synthesizing plane geometry diagrams by optimizing SDF representation against symbolic mathematical constraints.

- We design a more accessible user workflow where natural language is translated by an LLM into a symbolic representation, eliminating the need for specialized languages required by previous tools.

- We achieve high efficiency through a batch-enabled optimization process, allowing for the simultaneous generation of multiple complex diagrams. Furthermore, our framework could easily extend to the analytic and 3D geometric.

- To demonstrate our method's ability to generate high quantity and quality outputs, we augmented the existing data by synthesizing 32 diagrams with diversity for each problem description in FormalGeo7k dataset, finally forming a large size of dataset with 224k samples.

- We achieve SOTA results on the GeoQA benchmark with 95.9% accuracy, demonstrating that high-quality diagram synthesis directly translates to superior geometric problem-solving capabilities.

## 2 RELATED WORK

### 2.1 PLANE GEOMETRY PROBLEM SOLVING

Early approaches to plane geometry problem (PGP) solving relied on manually defined rules applied to small datasets Seo et al. (2015; 2014), resulting in poor generalization. Neural network-based models, such as NGS Chen et al. (2021) and DPE-NGS Chen et al. (2021), introduced visual question answering and specialized program generation, but still struggle with coarse-grained diagram understanding Ning et al. (2023). Symbolic reasoning methods, including Inter-GPS Lu et al. (2021) and FormalGeo Zhang et al. (2023b), use complex rule-based systems to interpret formal languages parsed from diagrams and text. However, their performance is constrained by limited datasets and parameter sizes, and they do not produce natural language solutions. Recent advances in multi-modal large language models (MLLMs), such as G-LLaVA Gao et al. (2025) and Geo-LLaVA Xu et al. (2024), fine-tune based models Liu et al. (2023) to generate natural language solutions. These approaches benefit from data augmentation techniques (e.g., Geo170K via GPT-3.5 Gao et al. (2025)), but often prioritize textual information over the diversity and complexity of geometry diagrams Zhuang et al. (2024). While methods like GeoX Xia et al. (2025) and R-CoT Deng et al. (2024) have made progress in diagram understanding, generating accurate, diverse, and controllable geometric diagrams remains a fundamental challenge for advancing PGP solving.

### 2.2 PLANE GEOMETRY DIAGRAM SYNTHESIS

Plane geometry diagram synthesis is critical for evaluating problem-solving systems but is under-explored due to complexity Rombach et al. (2022). Current rule-based methods, including Geom-Verse Kazemi et al. (2024) and MAVIS Zhang et al. (2025), stitch predefined shapes along edges. However, this limits them from generating diagrams with inscribed relationships (e.g., a triangle in a circle). Despite R-CoT's Deng et al. (2024) advancements with inscribed elements, existing approaches remain constrained by predefined shapes and rules, preventing synthesis from textual requirements. Another approach is Penrose Ye et al. (2020), which synthesizes diagrams from a domain-specific language by solving a constrained numerical optimization problem. However, for problems in Euclidean geometry that demand high mathematical fidelity, its reliance on a general-purpose constraint solver can be less ideal. Our work differs fundamentally in its representation: GeoSDF leverages Signed Distance Fields to directly embed core geometric properties—like distance, angle, and relations—into the loss functions. This specialized approach results in a more stable and direct convergence to mathematically precise configurations.

## 3 METHODOLOGY

The GeoSDF framework is developed for synthesizing plane geometry diagrams and consists of four steps, as illustrated in Fig. 2. Each step is described in detail in the following sections.

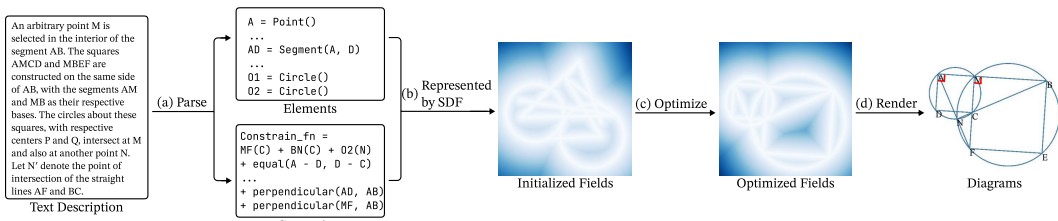

Figure 2: The GeoSDF Pipeline with an example IMO diagram synthesis process. (a) Parsing text description in natural language to geometric Elements and Constraints in symbolic language. (b) The elements and constraints are represented using SDF representations. (c) The optimization process from the initialized field to satisfy all the specified constraints forms the optimized Fields. (d) The final optimized fields are rendered to Diagrams that visualize the geometric problem.

Table 1: Predefined common constraints and loss functions in GeoSDF. In our framework, GeoSDF, we consider these geometric constraints directly as the loss functions to quantify the deviation from the desired configuration. The mathematical details are in Appendix C. Users could easily extend the extra loss function to satisfy other customized constraints.

| Category | Constraint | Loss Function | Description |
|---|---|---|---|
| Incidence | Equality | $\text{Equal}(A, B)$ | Two geometric elements $A, B$ have the same value. |
| | Less | $\text{Less}(A, B)$ | One numerical value is less than or equal to another. |
| Metric | Distance | $\text{Equal}(A(B), v)$ | A specific distance $v$ between two geometric elements $A$ to $B$. |
| | Angle | $\text{Angle}(A, B, C)$ | The angle formed by three points $A, B, C$ |
| | Area | $\text{Area}(A, ..., E)$ | A specific area for a polygon defined by a sequence of points. |
| Spatial | Parallelism | $\text{Parallel}(AB, CD)$ | Two lines or segments are parallel to each other. |
| | Perpendicularity | $\text{Perpendicular}(AB, CD)$ | Two line or segments are perpendicular to each other. |
| | Order | $\text{Order}(A, ..., E)$ | Counterclockwise order of points $A, ..., E$. |
| | On | $AB(C)$ | A specified point $C$ lies on a given segment or circle $AB$. |
| | Inside | $\text{Inside}(p, A, ..., E)$ | A point $p$ if it lies outside a convex polygon defined by other points $A$ to $E$. |
| Topological | Convexity | $\text{Convex}(A, ..., E)$ | A polygon formed by a sequence of points $A, ..., E$ is convex. |

### 3.1 PARSING NATURAL LANGUAGE INTO SYMBOLIC LANGUAGE

The first parsing step of the GeoSDF framework converts geometric natural language statements into a structured symbolic language representation in an end-to-end manner. As illustrated in Fig. 2, the symbolic language consists of **Elements**, which denote fundamental geometric objects such as points, lines, or segments, and **Constraints**, which describe relationships or properties among these objects. Following previous works Berg et al. (2008), we consider four types of constraints: (1) incidence (e.g., a point lies on a circle), (2) metric (e.g., distance between two points equals a given value), (3) spatial (e.g., two lines are parallel), and (4) topological (e.g., convexity is preserved), concluded in Table 1.

For instance, consider a case where two line segments are defined by points A, B and C, D, respectively, and the two segments are parallel. After processing through this step, the structured symbolic representation encodes the elements as `AB = SEGMENT(A, B)` and `CD = SEGMENT(C, D)`, while the constraint `Parallel(AB, CD)` captures the orientation relationship between the two segments. We fine-tune a QWen2.5-7B-Instruct model Qwen et al. (2025) on the training data in FormalGeo-7K Zhang et al. (2023b) to obtain the model for the parsing process.

### 3.2 REPRESENTATION BY SDF

We represent a **geometric diagram**, composed of the aforementioned **elements and constraints**, as a collection of signed distance fields (SDFs). An SDF is a continuous and differentiable scalar field that assigns to each point $(x, y) \in \mathbb{R}^2$ its shortest distance to a given geometric object. More details about SDF can be found in Appendix L.

Each **element** in the diagram is represented by its own SDF, which is fully determined by the element's continuous parameters (e.g., coordinates for points, radii for circles, lengths for segments). This representation captures the element's shape and position, providing a differentiable field that can later be optimized.

**Constraints** that encode geometric relationships between elements are expressed as differentiable functions of element parameters. They do not introduce new fields, rather, they specify conditions under which the combined element fields form a valid diagram. Each constraint thus acts as an optimization objective over the corresponding SDFs.

Finally, the complete diagram SDF is obtained by composing the individual element fields into a unified representation. As illustrated in Fig. 2, we begin by initializing a random field for each element. In the visualization, lighter colors indicate smaller SDF values, corresponding to points closer to the element. By sampling the field, we can observe the underlying structure of the initialized diagram. During optimization, the element parameters are adjusted to satisfy all constraints simultaneously, yielding a valid geometric configuration. We show a complete example in Appendix H.

### 3.3 Optimizing SDF Field

To clarify the optimization step, the **diagram** shows the complete set of $N$ geometric elements, $E = \{e_1, e_2, \ldots, e_N\}$. Each element $e_j \in E$ is associated with its own set of differentiable constraints $C_j = \{c_{j,k} \mid k = 1, 2, \ldots, M_j\}$, where $M_j$ is the number of constraints associated with element $e_j$, $k$ explicitly indexes the constraints for element $e_j$, so that $c_{j,k}$ denotes the $k$-th constraint in $C_j$.

The complete set of constraints across all elements is then $C = \bigcup_{j=1}^{N} C_j$. A valid geometric configuration is obtained when all constraints in $C$ are satisfied simultaneously. To enforce all geometric constraints, we have the loss function:

$$L_{constraints}(E, C) = \sum_{j=1}^{N} \sum_{c_{j,k} \in C_j} c_{j,k}(E), \tag{1}$$

where each $c_{j,k}$ is transformed into a non-negative differentiable function that evaluates to zero when the corresponding constraint is satisfied, ensuring the loss is bounded below by zero and suitable for gradient-based optimization.

To prevent elements from collapsing into overlapping or degenerate configurations, which could otherwise cause the optimization to fail, we introduce a crowd regularization term:

$$L_{crowd}(E) = \sum_{1 \leq i < j \leq N} \left[ \max\left(0, \tau_r - \|x_i - x_j\|\right) \right]^2, \tag{2}$$

where $x_i$ and $x_j$ denote the position parameters of elements $e_i$ and $e_j$, and the hyper-parameter $\tau_r$ specifies a distance between any pair of elements. $\tau_r$ is not sensitive; it only affects the distance among all elements.

The differentiable term can be optimized jointly with the SDF-based constraint losses. The total loss over all elements and constraints is then defined as:

$$L_{total} = L_{constraints}(E, C) + \lambda_{crowd} L_{crowd}(E), \tag{3}$$

where $\lambda_{crowd}$ balances the relative importance of the regularization. We provide a more detailed analysis for these hyperparameter in Appendix Q. This differentiable loss is minimized using standard gradient-based optimization, yielding a configuration $E^*$ in which constraints are approximately satisfied and elements remain well-spaced. Further visualization of the optimization procedure is provided in Appendix D.

### 3.4 Boundary Extraction and Visualization

The optimized configuration $E^*$ specifies the final shapes of all geometric elements. Each element's geometric parameters (e.g., center and radius of a circle) uniquely determine its boundary. Therefore, the boundaries can be directly computed from the elements representation $E^*$. Detailed boundary extraction procedures are provided in Appendix E.

## 4 Experiments

This section provides a comprehensive evaluation of the GeoSDF framework. We begin by detailing the implementation specifics 4.1. Next, we present qualitative synthesis results to demonstrate GeoSDF's capacity to generate diverse, geometrically consistent, and mathematically precise diagrams 4.2. We then quantify the performance of the natural-language-to-diagram parsing stage 4.3. Following the crucial property of quantifiability, which enables GeoSDF to directly solve PGPs by measuring properties from the synthesized diagram, achieving SOTA accuracy on the GeoQA benchmark 4.4. Finally, we include an ablation study and convergence analysis to examine the computational efficiency and the impact of the crowd regularization term on the synthesis process 4.6. Our method could be easily extended to the 3D diagram J.

### 4.1 Implementation Details

Our framework is implemented in Python using the PyTorch library and is structured as a user-friendly package for public release. For the optimization, we use the AdamW optimizer for a maximum of 10,000 iterations. The learning rate follows a cosine annealing schedule, decaying from an

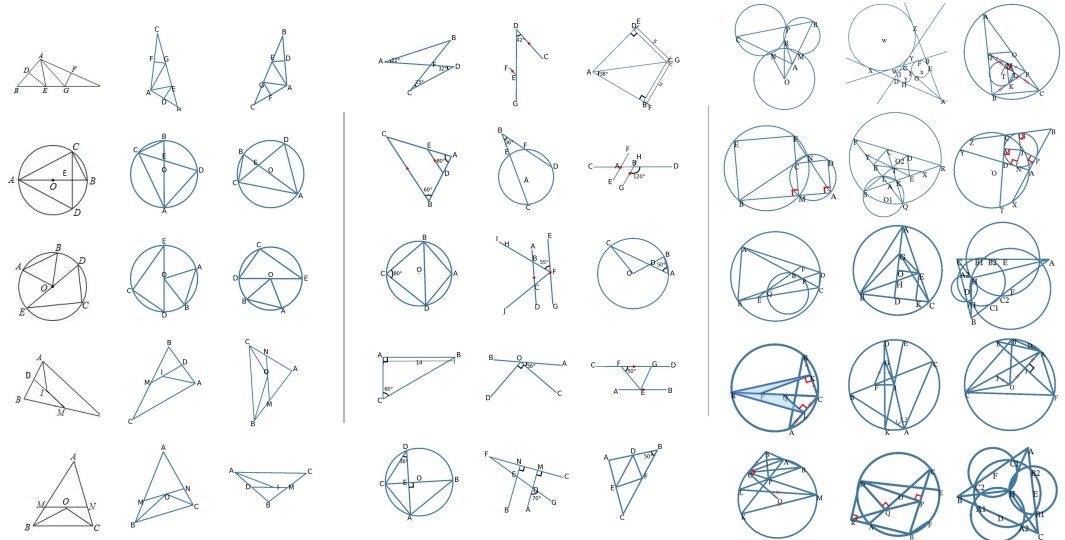

Figure 3: Qualitative results. Left: Synthesis on the FormalGeo7k dataset with diversity yet geometrically consistent (first column: ground truth). Middle: Synthesised with detailed annotations. Right: Synthesised IMO geometric problems with annotations.

initial value of 0.1 down to $1 \times 10^{-6}$. A synthesis is deemed successful if the total constraint loss falls below a threshold of 0.1. All hyperparameters are listed in the F

In our experiments, we select FormalGeo7k Zhang et al. (2023b) as the source datasets for providing the base geometry predicates as the input constraints of GeoSDF. FormalGeo7k consists of 6,981 PGPs, which were collected and re-annotated from previous PGP datasets, including GeoQA Chen et al. (2021), GeoQA-Plus Cao & Xiao (2022), and Geometry3K Lu et al. (2021). We leverage the annotated predicates from the FormalGeo7k to synthesize geometry diagrams using GeoSDF by first transforming these predicates into our structural and constraint-based representations and subsequently optimizing the SDF to generate precise and structurally valid geometry diagrams. For more challenging problems, we use the IMO problems selected by AlphaGeometry Trinh et al. (2024), for which we manually annotated the constraints from the problem statements.

## 4.2 QUALITATIVE SYNTHESIS RESULTS

**Geometry Diagram Synthesis.** We first applied GeoSDF to synthesize geometry diagrams based on the FormalGeo7k dataset. For each problem in the dataset, we leveraged the annotated textual predicates as the problem constraints and then used them to synthesise the corresponding plane geometry diagram. To demonstrate our method's ability to generate diverse outputs, we augmented the FormalGeo7k dataset by synthesizing 32 diagrams with diversity for each problem description from 7k to 224k diagrams. This process significantly increases the visual variety available for training and evaluation. A random selection of 400 synthesized examples is provided in the Appendix.

The visualization of five cases of these synthesized diagrams is shown in the left part of Figure 3. For each original problem, we select two new diagrams from the synthesized results, with the first row showing the ground truth and the other two representing the newly synthesized ones. The synthesized diagrams are geometrically consistent with the originals, but with rotation, mirror, and deformation diversity. Furthermore, our method provides a range of annotations for synthesized diagrams to enhance human readability, including parallelism, perpendicularity, lengths, areas, and angles. These annotations are clearly visualized in Figure 3 (Middle and Right).

**IMO Geometry Diagrams Synthesis.** Our GeoSDF can synthesize geometrically complex diagrams, particularly those found in International Mathematical Olympiad (IMO) problems. Previous methods, limited by their reliance on rule-based algorithms, which unable to generate such diagrams. Meanwhile, the model-based methods struggle with the deterministic rules and mathemat-

ical validity, resulting in synthesizing inaccurate diagrams. To demonstrate our method's ability to handle these challenges, we visualize the synthesis of IMO problems (from 1959 to 2022) in Figure 3 (right). These examples showcase our framework's ability to deal with diverse and challenging constraints. The resulting diagrams are both visually coherent and mathematically precise, demonstrating the potential of GeoSDF in handling sophisticated geometry synthesis tasks.

To verify the correctness of our synthesized IMO diagrams, we conducted a human evaluation study. We invited 20 participants with bachelor's degrees in science to assess whether the diagrams generated by GeoSDF are geometrically equivalent to the original diagram (i.e., ground truth). A total of 30 IMO geometry problems were used in the evaluation. On average, participants judged that **88.67%** (26.6 out of 30) of the synthesized diagrams fully reflected the geometry information in the original problem diagram. This result highlights that GeoSDF exhibits strong capability in synthesizing diagrams for IMO-level problems. The design of the questionnaire is included in Appendix K.

### 4.3 DIRECTLY GENERATING DIAGRAMS FROM NATURAL LANGUAGE

We assess the natural-language–to-diagram stage by evaluating the parser that converts natural problem text into symbolic constraints. Constraints are compared as order-independent sets, with F1 and Jaccard similarity used to measure overlap while balancing false positives and negatives Manning (2008). Before scoring, we apply light preprocessing: standardizing text (e.g., lowercasing, sorting arguments) and tokenizing while preserving expressions such as Collinear(A,B,C). All evaluations are conducted on the GeoQAChen et al. (2021) test set. The parser achieves solid overall performance with an F1 of **87.74%** and Jaccard of **83.53%**, showing that most constraints are captured reliably. These results suggest that the symbolic representation produced from natural text is able to support subsequent SDF-based optimization. While occasional parsing errors remain, they rarely alter the overall structure of the diagram, indicating the feasibility of the pipeline. We further provide a complete parsing example in Appendix I.

### 4.4 QUANTIFIABILITY OF GEOSDF

One of the significant properties of GeoSDF is the quantifiability. It means that all elements of the geometry diagram can be directly quantified or measured (e.g. degree of angle) and extracted (area and length will be scaled through the scale bar of each case). For example, the first problem in Fig. 4 asks for the measure of angle DFB (marked by the red arrow). Once the SDF optimization is complete and the diagram meets the loss constraint, we extract the angle value as the solution, which is 55. Similarly, in another example from Fig. 4, we determine the length of line YZ, the diameter of circle M. This capability enables us to extend its application to a broader range of experiments, such as direct problem solving and verifying whether the synthesized diagrams precisely satisfy the problem.

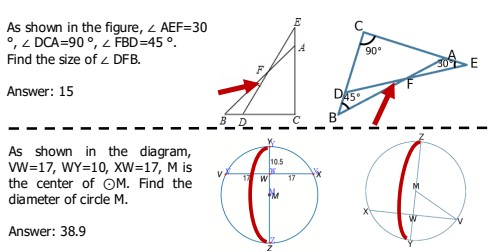

Figure 4: The example of quantifiability of GeoSDF. Our method is able to measure all the values in the diagram (e.g., degree, area), where the left diagram is the original diagram and the right one is synthesised by GeoSDF.

#### 4.4.1 PLANE GEOMETRY PROBLEM SOLVING

A key application of our framework is solving Plane Geometry Problems (PGPs) through a novel "solve-by-construction" paradigm. GeoSDF finds solutions by first synthesizing a high-fidelity diagram that satisfies all given geometric constraints and then directly measuring the properties of the target element (e.g., an angle's degree or a segment's length) from the final SDF representation.

We evaluated on the GeoQA Chen et al. (2021) dataset, comparing its performance against SOTA neural and MLLM-based solvers. To provide a comprehensive analysis, we test our method in three distinct settings, with results presented in Table 2. GeoSDF-GT (Oracle Input): This is our main result. We use the ground-truth symbolic constraints provided in the dataset to synthesize the diagram.

This setting isolates the performance of the core synthesis and measurement pipeline, removing any noise from upstream parsing errors. GeoSDF (End-to-End): To simulate a real-world scenario, this setup uses a fine-tuned LLM to parse the problem's text and diagram image into symbolic constraints, which are then fed into our solver. This evaluates the entire end-to-end pipeline.

Our primary method, GeoSDF, achieves a remarkable 94.5% accuracy in the completion setting and 95.9% in the choice setting. This result significantly surpasses all existing solvers, which perform logical or numerical reasoning. This highlights the potential of the solve-by-construction approach for a large part of geometry problems where the answer is a measurable quantity. Even in the end-to-end setting (GeoSDF), our method achieves SOTA accuracy of 78.5%, demonstrating its practical viability despite the challenges of automated parsing. These experiments underscore the power of high-precision diagram synthesis as a direct path to problem-solving. Furthermore, GeoSDF's quantifiable nature makes it a promising tool for external solution verification or as a reasoning module within hybrid PGP solvers. A detailed analysis of failure cases is provided in Appendix G.

### 4.4.2 SYNTHESISED DIAGRAM STRUCTURAL EQUIVALENCE CHECK

When using GeoSDF to synthesize diagrams based on known problem statements, questions, and answers, the quantifiability of GeoSDF enables further verification of whether the synthesized diagrams satisfy the given conditions and correspond to the correct answers. If the values extracted from the SDF representation align with the expected solution and the optimization loss remains below a predefined threshold, we consider the diagram a faithful representation of the original problem. This verification process uses the problem's goal formula to locate and extract relevant geometric features from the SDF object. To quantify this, we evaluated the success rate of diagram synthesis on the FormalGeo7K dataset. Among 6,981 samples, 5,943 diagrams (**85.13%**) passed the loss constraint check, and 5,697 diagrams (**81.60%**) successfully meet the target equivalence conditions. Upon evaluating the dataset, we identified that the main source of synthesis failures involves problems that are not strictly value-based. These cases typically include undefined

Table 2: Accuracy (%) on the GeoQA test set, the results are reported for two settings: **Completion**, the answer value is directly outputted; **Choice**, the problem choice is directly outputted. For neural PGP solvers, a random choice is selected if the completion result doesn't match any given choice.

| Model | Completion | Choice |
|---|---|---|
| NEURAL PGP SOLVERS | | |
| NGS Chen et al. (2021) | 60.0 | 69.5 |
| DPE-NGS Cao & Xiao (2022) | 62.7 | 70.4 |
| SCA-GPS Ning et al. (2023) | 64.1 | 72.7 |
| CLOSED SOURCE MLLMs | | |
| GPT-4o | - | 61.4 |
| Gemini-2.5 | - | 63.4 |
| MLLM PGP SOLVERS | | |
| G-LLaVA-13B Gao et al. (2025) | - | 67.0 |
| GeoX Xia et al. (2025) | 54.9 | - |
| Qwen2.5-VL-7B Bai et al. (2025) | 64.2 | 64.3 |
| InternVL2.5-8B Chen et al. (2024) | 50.5 | 58.1 |
| MAVIS-7B Zhang et al. (2025) | - | 68.3 |
| R-CoT-8B Deng et al. (2024) | - | 75.1 |
| GeoGen-SFT-7B Pan et al. (2025) | 64.6 | 78.0 |
| GeoUni-1.5B Cheng et al. (2025) | 66.7 | 78.0 |
| GeoSDF | 78.5 | 83.2 |
| GeoSDF-GT | **94.5** | **95.9** |

variables within their geometric constraints, such as defining a line segment's length as 'x+2'. While our parsing module capably translates these statements into symbolic constraints, the GeoSDF synthesis process cannot render a diagram without concrete numerical values.

### 4.5 EQUIVALENCE OF SYNTHESISED GEOSDF DIAGRAMS

To further assess whether the diagrams synthesized by GeoSDF preserve the same geometry information in the original diagrams, we conducted an experiment by replacing the geometry diagrams in the test set of GeoQA with newly synthesized versions. We then evaluated whether baseline models could still solve the problems using these new diagrams. We tested several baseline models, including NGS Chen

Table 3: Comparison (%) on the GeoQA test set between original diagrams and those synthesized by GeoSDF, evaluated across different base models.

| Test Set | Original Test Set | Synthesised by GeoSDF |
|---|---|---|
| NGS Chen et al. (2021) | 60.9 | 61.5 |
| SCA-GPS Ning et al. (2023) | 64.1 | 64.9 |
| G-LLaVA-7B Gao et al. (2025) | 64.2 | 64.3 |

et al. (2021) and G-LLaVA-7B Gao et al. (2025), using both the original and GeoSDF-synthesized test sets. All models were evaluated with their publicly released code, and model weights were trained on the original training set. To create the new diagrams, we annotated problem predicates and used GeoSDF to synthesize diagrams for each test example. The results of this experiment

are shown in Table 3. Interestingly, the accuracy on the GeoSDF-synthesized test set was slightly higher than on the original test set. We attribute this improvement to the fact that some of the original diagrams were of relatively low quality, which likely hindered model performance. In contrast, the diagrams synthesized by GeoSDF offer clearer visual representations, helping the models better understand the geometry and ultimately improving accuracy.

### 4.6 ABLATION STUDY

#### 4.6.1 COMPUTATIONAL EFFICIENCY AND CONVERGENCE ANALYSIS

The computational efficiency and convergence behavior of our GeoSDF framework are critical for its practical applicability. To evaluate computational performance, we compare the performance, resource usage, and time versus Batch Size for the IMO level figure as shown in Tab. 4. GeoSDF could also run on the CPU or the laptop platform due to it is not sensitive to the computing power. The optimization time is nearly constant versus Batch Size, and we find the bottleneck is in the transfer speed between the GPU and CPU. We measure the average optimization time for generating diverse geometric figures. For typical complex IMO level figures

Table 4: Performance, Resource Usage, and Time vs. Batch Size over IMO Diagram Synthesis. The Successful Rate is calculated as the loss of less than 0.1 in a batch. The Memory is the maximum GPU memory allocated throughout the entire process. We take 10000 steps for optimization.

| Batch Size | Successful Rate (%) | Memory (GB) | Time (s) |
|---|---|---|---|
| 2048 | 57.67 | 14.49 | 98.7 |
| 1024 | 57.42 | 7.02 | 60.5 |
| 512 | 56.84 | 3.52 | 42.7 |
| 256 | 60.16 | 1.77 | 33.6 |
| 128 | 60.94 | 0.89 | 28.8 |
| 64 | 62.19 | 0.50 | 26.5 |
| 32 | 53.13 | 0.38 | 25.3 |

(e.g., those involving 20+ elements and 20+ constraints), the generation process converges in about 20s with annotations. For example, the optimization time is about 21.3s for IMO 2021 P4 for all batch size configurations with 24 elements and 24 constraints. This performance demonstrates the feasibility of our approach for interactive or near-interactive geometric design.

#### 4.6.2 IMPACT OF CROWD REGULARIZATION

We conducted an ablation study to show the contribution of the crowd regularization term (i.e., Eqn. 2) in GeoSDF. This term aims to prevent geometric elements from collapsing, encouraging well-distributed diagrams. Without the crowd term, diagrams generated on the GeoQA dataset may degrade, with points overlapping and lines merging. Quantitatively, this resulted in a decrease of **3.7%** in the success rate of diagram synthesis on the FormalGeo7K dataset, ranging from 77.89% to 81.60%. This demonstrates that the crowd term enforces geometric separation and visual clarity.

#### 4.6.3 SENSITIVITY ANALYSIS

In this section, we analyze the sensitivity of the key hyperparameters: the crowd regularization threshold $\tau_r$, the crowd regularization weight $\lambda_{crowd}$, and the convergence loss threshold $\tau_c$. Furthermore, we evaluate the impact of Batch Size $B$ on performance. The final answer is computed as the average of the values from the subset of samples within the batch that satisfy the convergence loss threshold $\tau_c$. We adopted a univariate ablation strategy, varying one hyperparameter at a time while fixing the others to their default values (highlighted in bold in Table 5). All experiments are conducted on the GeoQA dataset utilizing the GeoSDF-GT configuration, as detailed in Section 4.4.2.

Table 5: Hyperparameter Sensitivity Analyze to crowd regularization threshold $\tau_r$, crowd regularization weight $\lambda_{crowd}$, convergence threshold of loss $\tau_c$, and the batch size $B$. The bold value is the default for all experiments. (acc. stand for Accuracy).

| | Value | Acc. | | Value | Acc. |
|---|---|---|---|---|---|
| $\tau_r$ | 0.05 | 94.6 | $\lambda_{crowd}$ | 1.0 | 94.9 |
| | **0.2** | **94.5** | | 5.0 | 94.5 |
| | 1 | 94.5 | | **10.0** | **94.5** |
| | 5 | 94.4 | | 20.0 | 94.3 |
| $\tau_c$ | 0.01 | 92.5 | $B$ | 1 | 70.4 |
| | 0.03 | 94.1 | | 16 | 93.7 |
| | **0.1** | **94.5** | | **32** | **94.5** |
| | 1 | 94.8 | | 64 | 94.5 |

From the Table 5, the framework demonstrates stability across the tested hyperparameters. The crowd regularization weight $\lambda_{crowd}$ and threshold $\tau_r$ exhibit significant robustness; notably, varying $\lambda_{crowd}$ from 0.1 to 10.0 results in negligible fluctuations in accuracy, maintaining performance about $94.5\%$. In contrast, the convergence threshold

$\tau_c$ is critical to model performance. Relaxing $\tau_c$ to 1.0 precipitates a substantial decline in accuracy to 70.4%, as the looser constraint permits the acceptance of geometrically inaccurate diagrams that have not fully converged. A comprehensive table of all hyperparameters is provided in Appendix F.

### 4.7 EXTENSIONS TO ANALYTIC AND 3D GEOMETRY

A distinguishing feature of the GeoSDF framework is its inherent extensibility. Our SDF-based approach allows for the inclusion of arbitrary geometric shapes provided their distance functions can be defined differentiably, ranging from conic sections to 3D solids. In this section, we demonstrate the applications on Analytic Geometry and 3D Geometry in GeoSDF.

**Analytic Geometry and Conic Sections.** The synthesis of analytic geometry diagrams requires handling curves defined by implicit equations rather than simple constructive steps. We extend our element set to include conic sections, such as hyperbolas and spline curves.

**3D Geometry Synthesis.** The vector-based formulation of GeoSDF naturally generalizes from the 2D plane ($\mathbb{R}^2$) to 3D space ($\mathbb{R}^3$). The optimization pipeline remains structurally identical, as the definition of constraints such as distance, parallelism, and perpendicularity are dimension-agnostic.

As shown in the experimental results in figure 5, the framework successfully satisfies 3D spatial constraints demonstrating that GeoSDF is not limited to planar diagrams but is a robust tool for multi-dimensional geometric analysis. Please see more analysis in Appendix B and J.

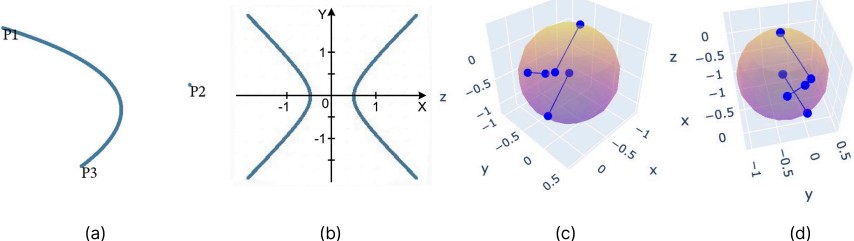

Figure 5: Extensions to Analytic and 3D Geometry. (a) Quadratic Bézier Curve, P2 is the control point, (b) Hyperbola with the 2D axis, (c) and (d) are two point of viewers in the 3D scene to meet the parallel and perpendicular requirements on a sphere.

## 5 CONCLUSION

In this work, we introduced GeoSDF, a novel approach for synthesizing plane geometry diagrams by leveraging Signed Distance Field. GeoSDF can generate precise and semantically aligned diagrams, supported by an integrated self-verification mechanism that ensures consistency with the original problem constraints. By leveraging the implicit representation and differentiability of SDFs, GeoSDF enables flexible and accurate diagram synthesis. Our extensive experiments demonstrate that GeoSDF can produce high-quality diagrams, preform well across a wide range of problems, including complex International Mathematical Olympiad (IMO) problems, and effectively handle intricate geometric relationships and constraints. A key advantage of GeoSDF is its quantifiability, which opens up promising applications such as directly solving geometry problems. Experimental results show that GeoSDF significantly outperforms existing SOTA solvers, highlighting its strong potential to contribute to the field of solving geometry problems. Meanwhile, GeoSDF also maintains strong computational efficiency, making it well-suited for deployment across various application domains with minimal computational resource requirements. Our GeoSDF bridges the gap between computational graphics and AI4Math, offering a robust tool for advancing geometric reasoning in MLLMs and beyond. Future work will explore GeoSDF to handle more types of geometry diagrams, including 3D geometry and analytic geometry.

## 6 ETHICS STATEMENT

We have read and adhered to the ICLR Code of Ethics. Our research focuses on the automated synthesis of mathematical diagrams, a fundamental task in computational geometry and AI for education. We have considered the ethical implications of our work and foresee no direct negative societal consequences.

The datasets used in our research, FormalGeo7k and the IMO problem set, are established public benchmarks in the academic community. Our contribution includes a new dataset of synthesized diagrams, which is derived from these public sources and contains no personally identifiable or sensitive information. We believe that our tool, GeoSDF, serves a beneficial purpose by providing a means to generate high-quality educational materials and to advance research in automated mathematical reasoning.

## 7 REPRODUCIBILITY STATEMENT

To ensure the reproducibility of our work, we commit to making our code, data, and experimental setup fully available.

Code and Tool: The complete source code for the GeoSDF framework, along with the scripts required to reproduce all experimental results presented in this paper, will be made publicly available in a GitHub repository after peer review. We will also release the user-friendly online tool mentioned in the abstract.

Datasets: The experiments are based on the publicly available FormalGeo7k and IMO datasets. Furthermore, we will release the complete set of diagrams synthesized by GeoSDF for our experiments, allowing for direct verification of our results and reuse by the community.

Experimental Details: A detailed description of the implementation, hyperparameters, and optimization settings can be found in Section 4.1. Further analysis, including the failure analysis and additional results that provide deeper insight into the method's behavior, are located in the Appendix.

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

# A GEOMETRIC ELEMENTS

We represent the scene geometry using a collection of differentiable distance functions. For a given query coordinate $\mathbf{p} \in \mathbb{R}^2$, each primitive defines a scalar field $f(\mathbf{p}) : \mathbb{R}^2 \to \mathbb{R}_{\geq 0}$ representing the Euclidean distance from $\mathbf{p}$ to the closest point on the geometry. We employ Signed Distance Functions for all primitives.

## A.1 POINT AND CIRCLE PRIMITIVES

The simplest primitive is a 0-dimensional point defined by a learnable center $\mathbf{c}$. The distance function is given by the Euclidean norm:

$$f_{\text{point}}(\mathbf{p}; \mathbf{c}) = \|\mathbf{p} - \mathbf{c}\|_2. \tag{4}$$

For a circular primitive with center $\mathbf{c}$ and radius $r$, both of them are learnable. The distance is defined as the absolute difference between the distance to the center and the radius. The use of the absolute value ensures the function remains non-negative to keep the loss always non-negative:

$$f_{\text{circle}}(\mathbf{p}; \mathbf{c}, r) = \big| \|\mathbf{p} - \mathbf{c}\|_2 - r \big|. \tag{5}$$

## A.2 LINEAR PRIMITIVES

We model linear structures using two points, $\mathbf{a}$ and $\mathbf{b}$. Let $\mathbf{v}_{ab} = \mathbf{b} - \mathbf{a}$ be the vector along the line structure, and $\mathbf{v}_{ap} = \mathbf{p} - \mathbf{a}$ be the vector from the start point to the query point. The two points a and b are the learnable parameters. To determine the distance, we first compute the projection scalar $h$, which represents the normalized position of the closest point along the vector $\mathbf{v}_{ab}$:

$$h = \frac{\mathbf{v}_{ap} \cdot \mathbf{v}_{ab}}{\|\mathbf{v}_{ab}\|_2^2}. \tag{6}$$

**Infinite Line.** For a line extending infinitely in both directions, the projection $h$ is unconstrained. The distance is the magnitude of the rejection vector:

$$f_{\text{line}}(\mathbf{p}; \mathbf{a}, \mathbf{b}) = \|\mathbf{v}_{ap} - h \cdot \mathbf{v}_{ab}\|_2. \tag{7}$$

**Segment.** For a finite segment bounded by $\mathbf{a}$ and $\mathbf{b}$, the closest point must lie between the endpoints. We constrain the projection scalar $h$ to the interval $[0, 1]$ using a clamping operation:

$$\bar{h} = \text{clamp}(h, 0, 1) = \min(\max(h, 0), 1). \tag{8}$$

The distance is then computed using the clamped scalar $\bar{h}$:

$$f_{\text{segment}}(\mathbf{p}; \mathbf{a}, \mathbf{b}) = \|\mathbf{v}_{ap} - \bar{h} \cdot \mathbf{v}_{ab}\|_2. \tag{9}$$

## A.3 COMPOSITION

Complex shapes are constructed by composing multiple primitives. We implement the boolean Union operation, which corresponds to the minimum operator in the distance field domain. Given N distance fields $f_A, f_B, \ldots f_N$, the unified field is:

$$f_{\text{union}}(\mathbf{p}) = \min(f_A(\mathbf{p}), f_B(\mathbf{p}), \ldots, f_N(\mathbf{p})). \tag{10}$$

## B  COMPLEX GEOMETRIC ELEMENTS

The GeoSDF framework is naturally extensible. Unlike rule-based constructive geometry, adding a Conic Section (e.g., a Hyperbola), spline curves, and freeform loci only requires defining its differentiable distance function.

**Hyperbola.** The Hyperbola implements the distance to the curve defined by the implicit equation:

$$x \cdot y = k.$$

The input point $\mathbf{p}$ is first transformed via a 45-degree rotation and symmetry operations (using absolute values) to map the general hyperbola shape into this standard form. The geometric transformation applied is:

$$u = \frac{|p_x| - |p_y|}{\sqrt{2}}, \quad v = \frac{|p_x| + |p_y|}{\sqrt{2}}$$

The optimization problem seeks a point $\mathbf{q} = (t, k/t)$ on the curve that minimizes the distance to the transformed point $(u, v)$. This results in minimizing:

$$\|(u, v) - (t, k/t)\|^2$$

Taking the derivative with respect to $t$ leads to a quartic equation (degree 4), which is solved analytically to find the closest point. The signed distance is given by:

$$d(\mathbf{p}) = \text{sign}(k - u \cdot v) \cdot \min_{\mathbf{q} \in \text{Curve}} \|\mathbf{p}_{rot} - \mathbf{q}\|$$

The learnable parameters are $k$ and $he$. $k$ is the hyperbola constant. It determines the "width" or distance of the vertex from the origin. For $xy = k$, the vertex is at $(\sqrt{k}, \sqrt{k})$. $he$ represents the height extent. This parameter is used to calculate a minimum value for $t$, effectively cutting off the hyperbola's arms. It turns the infinite hyperbola into a finite segment.

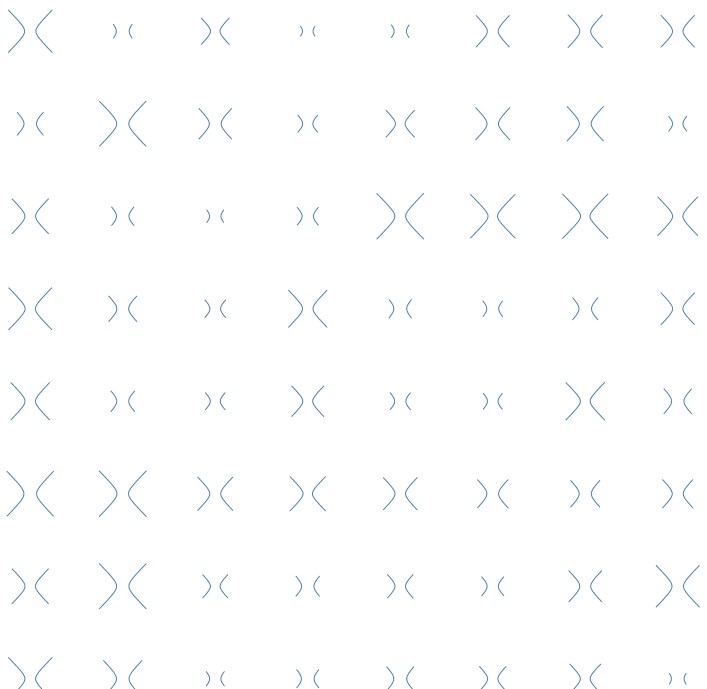

Figure 6: A visualization of a Hyperbola in a batch.

**Quadratic Bézier Curve.** A quadratic Bézier curve is defined parametrically by three learnable control points $\mathbf{P}_0, \mathbf{P}_1, \mathbf{P}_2 \in \mathbb{R}^2$ and a parameter $t \in [0, 1]$. $\mathbf{P}_0$ and $\mathbf{P}_2$ are the Start and End point coordinates. $\mathbf{P}_1$ is the Control point coordinates (determines curvature).

$$\mathbf{B}(t) = (1-t)^2\mathbf{P}_0 + 2(1-t)t\mathbf{P}_1 + t^2\mathbf{P}_2$$

This is often rearranged for optimization as:

$$\mathbf{B}(t) = \mathbf{P}_0 + 2t(\mathbf{P}_1 - \mathbf{P}_0) + t^2(\mathbf{P}_0 - 2\mathbf{P}_1 + \mathbf{P}_2)$$

The distance function $d(\mathbf{p})$ for a query point $\mathbf{p}$ is defined as the minimum distance to the curve segment:

$$d(\mathbf{p}) = \min_{t \in [0,1]} \|\mathbf{p} - \mathbf{B}(t)\|$$

To find the optimal $t$, we minimize the squared distance $f(t) = \|\mathbf{p} - \mathbf{B}(t)\|^2$. Setting the derivative $f'(t) = 0$ results in a cubic equation in $t$. We use Cardano's method to solve the cubic equation analytically to find the roots, clamp them to the range $[0, 1]$, and compute the distance.

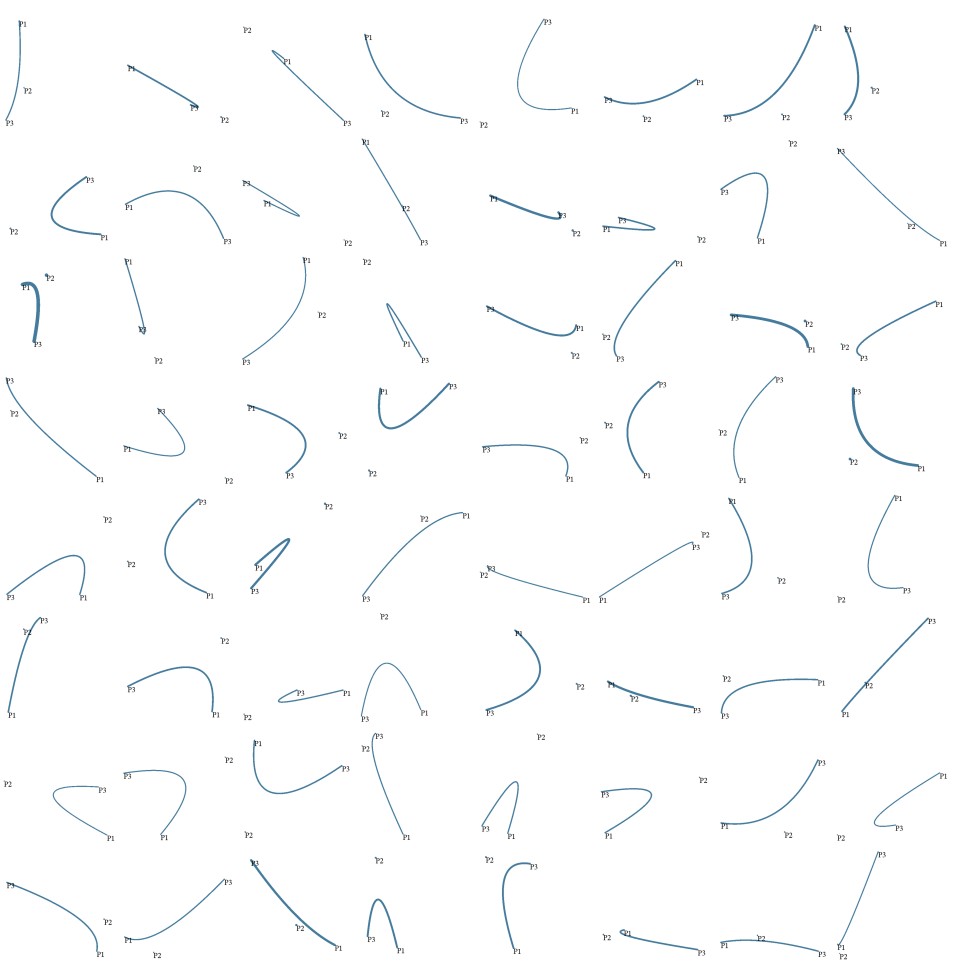

Figure 7: A visualization of a Quadratic Bezier in a batch.

## C  CONSTRAINTS DETAILS IN MATHEMATIC

**Equal.** Computes the absolute difference between two values, $A$ and $B$:

$$\text{Equal}(A, B) = |A - B|$$

**Less.** A differentiable version of the less-than-or-equal constraint.

$$\text{less}(A, B) = \max(A - B, 0)$$

This can also be written as:

$$\text{less}(a, b) = \begin{cases} 0 & \text{if } a \leq b \\ a - b & \text{if } a > b \end{cases}$$

**Angle.** For three points $A$, $B$, and $C$ (with $B$ as the vertex), the angle $\theta$ at $B$ is computed using the dot product of vectors $\vec{BA}$ and $\vec{BC}$:

$$\text{Angle}(A, B, C) = \arccos\left(\frac{\vec{BA} \cdot \vec{BC}}{\|\vec{BA}\|\|\vec{BC}\|}\right),$$

where $\vec{BA} = A - B, \vec{BC} = C - B$.

**Area.** Given vertices $(x_1, y_1), (x_2, y_2), \ldots, (x_n, y_n)$ of a polygon (assuming $(x_{n+1}, y_{n+1}) = (x_1, y_1)$), the area is:

$$\text{Area}((x_1, y_1), \ldots, (x_n, y_n)) = \frac{1}{2}\left|\sum_{i=1}^{n} (x_i y_{i+1} - y_i x_{i+1})\right|$$

**Parallel.** For two line segments $AB$ and $CD$, let their direction vectors be

$$v_1 = B - A, \quad v_2 = D - C.$$

In 2D, the cross product (a scalar) that measures the non-parallelism is:

$$L_{\text{parallel}} = |v_{1x}v_{2y} - v_{1y}v_{2x}|$$

Next, define the minimum distance between the segments as:

$$d = \min\{d(A, \overline{CD}), \ d(B, \overline{CD}), \ d(C, \overline{AB}), \ d(D, \overline{AB})\}$$

A distance penalty is applied if this distance is below a threshold $T$ (with weight $\lambda_{parallel}$) to avoid the elements crowd:

$$P_{\text{distance}} = \lambda_{parallel} \max\{0, T - d\}$$

Thus, the final constraint is:

$$\text{Parallel}(AB, CD) = |v_{1x}v_{2y} - v_{1y}v_{2x}| + \lambda_{parallel} \max\{0, T - d\}$$

**Perpendicular.** When the lines are defined by endpoints $A_1 A_2$ and $B_1 B_2$, let:

$$v_1 = A_2 - A_1, \quad v_2 = B_2 - B_1.$$

The perpendicularity constraint remains:

$$\text{Perpendicular}(AB, CD) = |v_1 \cdot v_2|$$

**Order.** For points on a circle with center $O$, let the angle from $O$ to each point be $\theta_i$ (after unwrapping so they are continuous). The difference between consecutive angles is:

$$\Delta\theta_i = \theta_{i+1} - \theta_i$$

A penalty is applied when a difference is too small (below a small constant $\epsilon$):

$$\text{Order}((x_1, y_1), \ldots, (x_n, y_n)) = \sum_{i=1}^{n-1} \max\{0, \epsilon - \Delta\theta_i\}$$

**Inside.** Determines whether a point lies inside a convex polygon. Given a point $P$ and a convex polygon defined by vertices $V_1, V_2, \ldots, V_n$ ordered either clockwise or counterclockwise. For each edge of the polygon, we define vectors:

$$\vec{e_i} = \vec{V}{i+1} - \vec{V_i} \quad \text{for } i = 1, 2, \ldots, n,$$

where $\vec{V}_{n+1} = \vec{V_1}$ (to close the polygon) For each vertex, we compute vectors from the vertex to point $P$:

$$\vec{d_i} = \vec{P} - \vec{V_i} \quad \text{for } i = 1, 2, \ldots, n,$$

Calculate the cross products between edge vectors and vectors to point $P$:

$$c_i = \vec{e_i} \times \vec{d_i} = (e{i,x} \cdot d{i,y} - e_{i,y} \cdot d_{i,x}) \quad \text{for } i = 1, 2, \ldots, n.$$

For a point inside the polygon, all cross products should have the same sign. Let $s_1 = \text{sign}(c_1)$ be the sign of the first cross product. The penalty for each edge is defined as:

$$p_i = \max(0, -s_1 \cdot c_i) \quad \text{for } i = 1, 2, \ldots, n.$$

The total penalty is the sum of all individual penalties:

$$\text{penalty} = \sum_{i=1}^{n} p_i$$

If penalty $= 0$, then point $P$ is inside the convex polygon. Otherwise, $P$ is outside the polygon.

**Crowd Penalty.** Given $N$ points $x_1, x_2, \ldots, x_N \in \mathbb{R}^D$, define the Euclidean distance between points $x_i$ and $x_j$ as:

$$d_{ij} = \|x_i - x_j\|$$

The penalty for each pair (only considering $i < j$) is:

$$p_{ij} = \max\{0, \tau_r - d_{ij}\}^2$$

The Crowd Penalty is:

$$P_{\text{Crowd}} = \sum_{1 \leq i < j \leq N} \max\{0, \tau_r - d_{ij}\}^2,$$

where the $\tau_r$ is the hyper-parameter to control the distance.

We make this algorithm vectorized to speed up the calculation on the GPU.

# D  OPTIMIZATION PROCESS VISUALIZATION

We visualized the process of the optimization of the GeoSDF in Fig. 8 with four IMO examples of the geometry diagrams. The leftmost diagrams represent the initial state, while the rightmost diagrams depict the final state after the optimization process. The initial state is set randomly. Through optimization, the SDF diagrams gradually evolve from a random configuration $E$ to $E^*$, ultimately satisfying the diagram's constraints.

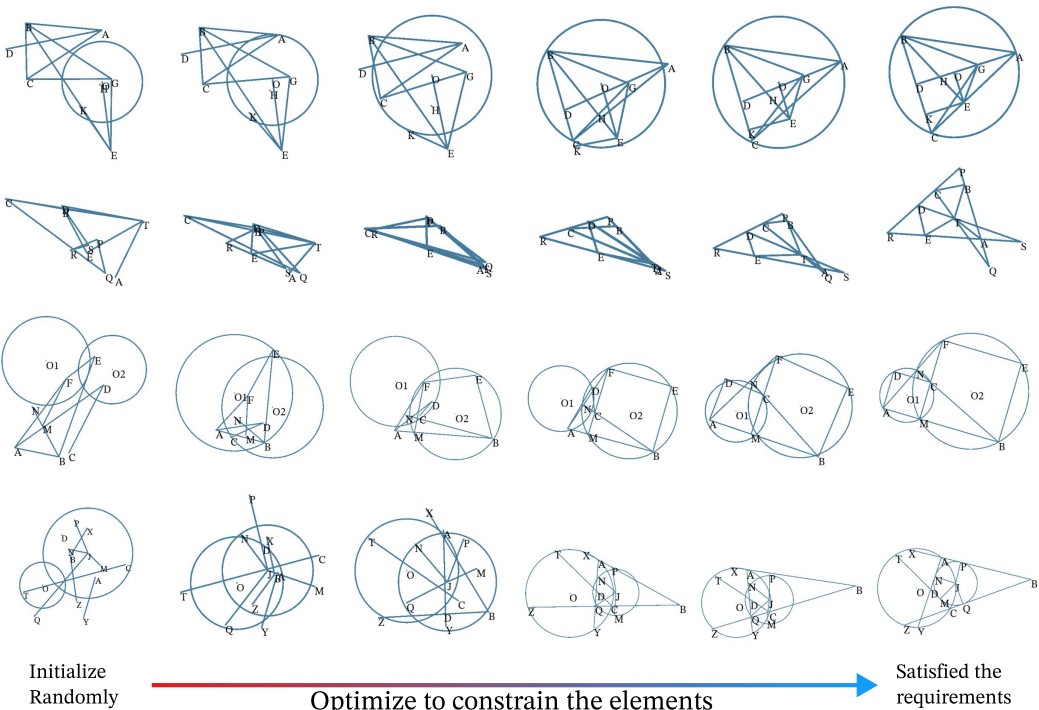

Initialize Randomly

Optimize to constrain the elements

Satisfied the requirements

Figure 8: The visualization of the optimization process. We first randomly initialize the geometry elements and then optimize the SDF by geometry constraints to the synthesized geometry diagram. The upper two examples are selected from FormalGeo-IMO Zhang et al. (2023b), and the lower two examples are from IMO 1959 Question 5 and IMO 2021 Question 4.

# E  BOUNDARY THRESHOLD

The optimized configuration $E^*$ allows for sampling the SDF of the geometric elements and evaluating the zero iso-surface to render the corresponding point set. To achieve this, a $N^2$ pixel grid is sampled. A threshold $\tau_t$ is then applied to the SDF values to extract the boundaries of the geometric shapes. This threshold determines the sampling range for the field, and its impact on the visualization is presented in Figure 9. Different threshold values result in varying levels of detail in the geometric diagram; specifically, a higher threshold expands the element boundaries, leading to thicker lines.

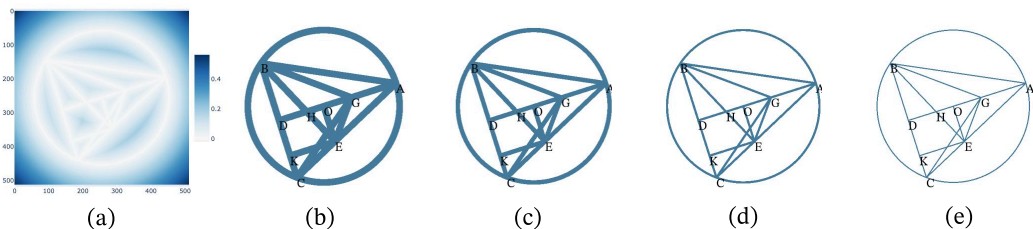

| (a) | (b) | (c) | (d) | (e) |

Figure 9: (a) The values sampled from SDF by $512 \times 512$ pixel grid visualization. And visualizations of its zero iso-surface with different thresholds (b) 0.3, (c) 0.2, (d) 0.1, (e) 0.005.

# F  HYPER-PARAMETERS

Here we list all hyperparameters mentioned before in our method in table F.

Table 6: Hyperparameters and default values used in GeoSDF optimization and constraints.

| Symbol | Description | Default Value |
|---|---|---|
| *Optimization Settings* | | |
| $N_{steps}$ | Maximum optimization iterations | $10,000$ |
| $lr$ | Learning rate (Cosine Annealing) | $0.1 \rightarrow 1 \times 10^{-6}$ |
| $Optimizer$ | Optimization algorithm | AdamW |
| $B$ | Batch size 4 | $16 \rightarrow 1024$ |
| $L_{thresh}$ | Convergence success threshold (Loss) 4.6.1 | 0.1 |
| *Loss Function & Regularization* | | |
| $\tau_r$ | Crowd regularization distance 2 | 0.2 |
| $\tau_t$ | Boundary thickness E | 0.1 |
| $\tau_c$ | Convergence constraint loss threshold 4.1 | 0.1 |
| $\lambda_{crowd}$ | Crowd regularization weight 3 | 3.0 |
| $\lambda_{parallel}$ | Parallelism distance penalty weight A | 0.3 |

# G  FAILURE ANALYSIS

While GeoSDF demonstrates strong performance, it is important to analyze its failure modes, which primarily fall into two categories: optimization challenges and parsing limitations.

**Optimization Failures.** The core of our method relies on solving a non-convex optimization problem. Consequently, the optimizer can sometimes converge to a local minimum or a saddle point that does not represent a valid geometric configuration. These failures manifest in two ways mainly:

(a) High Final Loss: This failure case occurs when the optimization process terminates without satisfying all constraints, resulting in a high total loss. This happens in highly constrained problems where the optimization is particularly complex. We treat these cases as synthesis failures. To deal with this type of failure, a convergence threshold is set to filter the failure cases out. In the implementation, all the diagrams that loss above our convergence threshold of $0.1$ are dropped out.

(b) Geometric Degeneracies: In some instances, the optimizer might produce a diagram with a low loss, but which is still visually or geometrically incorrect. This can include degenerate cases like three points becoming collinear when they should form a non-trivial triangle, or elements overlapping in a way that violates implicit assumptions of the problem. Our crowd regularization term (Eqn. 2) mitigates many of these issues, but they are not entirely eliminated.

**Natural Language Parsing Limitations.** The accuracy of the entire pipeline is fundamentally dependent on the initial NL-to-symbolic parsing step. We identified two primary sources of error from our fine-tuned LLM parser:

(a) Implicit information during LLM parsing: Many geometry problems contain implicit information that is not explicitly stated in the text. For example, a problem might refer to the intersection of two lines without explicitly defining a point at that intersection. Our current parser struggles to infer and create these necessary-but-unstated geometric elements, leading to an incomplete set of constraints for synthesis.

(b) Ambiguity and Errors: Ambiguous phrasing in the problem description can lead to incorrect symbolic translations. Furthermore, while our framework can detect syntactical errors (via the Python compiler) and logically conflicting constraints (via high residual loss on the conflicting terms), it cannot correct semantic mistakes by the user. For example, if the user misinterprets "tangent to" as "intersecting," the resulting diagram will be fundamentally flawed despite the optimization succeeding on the misinterpreted constraints. Diagnosing these failures is straightforward. By inspecting the per-constraint loss values, users can immediately identify which geometric relationships were not satisfied, providing clear feedback on the source of the optimization failure.

# H  EXAMPLES OF IMO FIGURE SYNTHESIZE PROCESS

In this section, we present a complete IMO geometric synthesis process implemented in Python. It should be noted that the initialization of elements and constraints can be executed automatically. These elements are maintained here for completeness.

The initial state figure will be synthesized as Fig. 10 with random parameters for each element. Then, define the constraints based on the mathematical geometric conditions. Finally, optimize the constraints and render the Fig. 11.

```python
from sdf2d import *
# the set of elements E
# batch size is 64
s = Shape(batch=64)
s.A = Point()
s.B = Point()
s.C = Point()
s.D = Point()
s.E = Point()
s.G = Point()
s.H = Point()
s.K = Point()
s.O = Point()
s.AB = Segment(s.A, s.B)
s.AC = Segment(s.A, s.C)
s.AD = Segment(s.A, s.D)
s.BC = Segment(s.B, s.C)
s.BE = Segment(s.B, s.E)
s.BG = Segment(s.B, s.G)
s.CG = Segment(s.C, s.G)
s.EG = Segment(s.E, s.G)
s.EK = Segment(s.E, s.K)
s.EO = Segment(s.E, s.O)
s.circle_O = Circle(center=s.O)
s.render() # sample the SDF

def constraints():
    return (
        s.AD(s.G)
        + s.AD(s.O)
        + s.AD(s.H)
        + s.BE(s.H)
        + s.BC(s.D)
        + s.BC(s.K)
        + s.AC(s.E)
        + s.circle_O(s.A)
        + s.circle_O(s.B)
        + s.circle_O(s.C)
        + equal(s.A - s.B, s.A - s.C)
        + equal(s.A - s.H,
          2*(s.D - s.circle_O))
        + equal(s.A - s.G, s.H - s.G)
        + parallel(s.EO, s.BC)
        + perpendicular(s.BE, s.AC)
        + perpendicular(s.BC, s.EK)
        + crowd_penalty(s.points())
    )
s.optimize(constraints)
s.render()
```

Figure 10: Initial state diagram in a batch, containing 64 samples.

Figure 11: Batched results in final state after optimization.

# I SYNTHESIS GEOMETRIC DIAGRAM FROM NATURAL LANGUAGE.

Here is an example of using a fine-tuned LLM to convert the natural language to the Diagram. The synthesized diagram is shown as Figure 12.

**Prompt:** Convert the given problem text into the Symbolic Language of Elements, Constraints, and the Goal. Problem Text: In the triangle ABC, AB=BC, D is the midpoint of AC. DE $\perp$ BC on E. The F is the midpoint of DE. G is the intersection point of AE and BF. Find the angle of AGB.

**Output:**

```
### Elements:
s = Shape(B=64)
s.A = Point()
s.B = Point()
s.C = Point()
s.D = Point()
s.E = Point()
s.F = Point()
s.G = Point()

s.AB = Segment(s.A, s.B)
s.BC = Segment(s.B, s.C)
s.AC = Segment(s.A, s.C)
s.AE = Segment(s.A, s.E)
s.BF = Segment(s.B, s.F)
s.DE = Segment(s.D, s.E)
```

```
### Constraints:
def constraints_fn(step):
    return (
        +s.BC(s.E)
        + s.AC(s.D)
        + s.AE(s.G)
        + s.BF(s.G)
        + s.DE(s.F)
        + eq(s.A - s.B, s.B - s.C)
        + eq(s.A - s.D, s.D - s.C)
        + eq(s.D - s.F, s.E - s.F)
        + perpendicular(s.DE, s.BC)
        + crowd_penalty(s.points_tensor())
    )

### Goal:
angle(s.A, s.G, s.B)
```

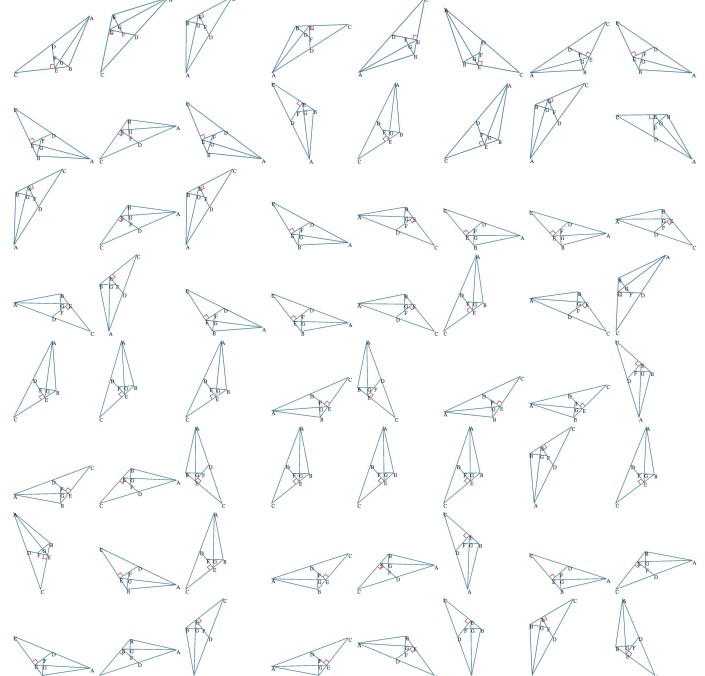

Figure 12: A batch of synthesised output diagrams by using natural language.

## J  3D GEOMETRIC DIAGRAMS

The GeoSDF is flexible to extend from the 2D diagrams to the 3D diagrams. Here we show a toy example. In 3D space, $ab$, $cd$, and $ef$ are segments. There exists a sphere with center $p$. Points $a$, $c$, $d$, $e$, and $f$ all lie on the surface of the sphere. Segment $ap$ is parallel to $cd$, and segment $ef$ is perpendicular to $cd$.

```python
from sdf3d import *
s = Shape(B=1)

s.c = Point3D()
s.d = Point3D()
s.cd = Segment3D(s.c, s.d)
s.e = Point3D()
s.f = Point3D()
s.ef = Segment3D(s.e, s.f)
s.a = Point3D()
s.p = Point3D()
s.ap = Segment3D(s.a, s.p)
s.sphere = Sphere(center=s.p)
s.render()
```

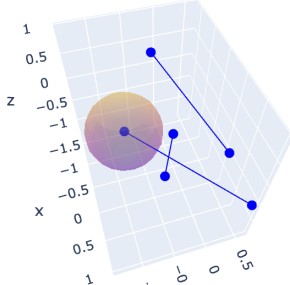

Figure 13: The 3D diagram initial state. We use Plotly to render the 3D diagram. The match cube method also works.

```python
def constraints_fn():
    return (
        # segment ap parallel to cd
        parallel(s.ap, s.cd)
        # segment ef perpendicular to cd
        + perpendicular(s.ef, s.cd)
        # Point a on the sphere
        + s.sphere(s.a)
        # Point c on the sphere
        + s.sphere(s.c)
        # Point d on the sphere
        + s.sphere(s.d)
        # Point e on the sphere
        + s.sphere(s.e)
        # Point f on the sphere
        + s.sphere(s.f)
    )
s.optimize(constraints_fn)
s.render()
```

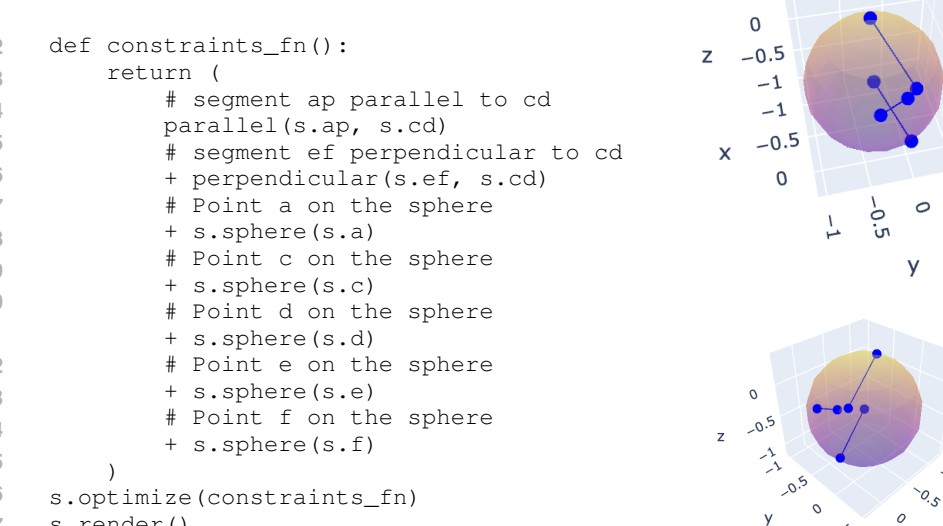

Figure 14: Optimized 3D diagram from two points of view.

## K    THE DESIGN OF THE QUESTIONNAIRE.

The Figure 15 is an example we designed for the questionnaire.

**Synthesized Image Logical Equivalence Evaluation**

**Instructions:**
For each pair of images (Ground Truth and Synthesis results), please answer the following questions. Focus on whether the synthesized image conveys the same fundamental meaning, objects, relationships, and context as the ground truth,

| Synthesis Results | Ground Truth | Answer |
| --- | --- | --- |
| | | ☐ Yes
☐ No |
| | | ☐ Yes
☐ No |
| | | ☐ Yes
☐ No |
| | | ☐ Yes
☐ No |

Figure 15: The design of the questionnaire.

## L    SIGNED DISTANCE FIELD

**Signed Distance Field Definition** A field in mathematics is a function that assigns a value (scalar or vector) to each point in a space. The space provides the coordinate system and the notion of distance necessary. The Signed Distance Field assigns a signed distance to the nearest point on the boundary of the shape for every point in this space. In mathematics, the SDF $F$ is a scalar field that assigns a value $s$ to every position $x$ in the space by

$$F(x) = s, \quad x \in \mathbb{R}^2, s \in \mathbb{R},$$

where the value $s$ indicates the distance from the position $x$ to the shape $\Omega \in \mathbb{R}^2$ in the two-dimensional plane space. The distance between a point $x$ and the boundary of the shape $B$ is hereby defined as

$$d(x, B) = \inf_{y \in B} d(x, y), \tag{11}$$

where $\inf$ denotes the infimum and $d$ represents the distance. For simplicity, that means the assigned value to the space corresponds to the minimum distance from the point to the geometric shape.

**SDF Related Work.** SDFs have been widely explored for their versatility in representing and rendering complex 3D geometries. Early foundational work by Frisken et al. Frisken et al. (2000) introduced Adaptively Sampled Distance Fields (ADFs), optimizing memory usage while preserving geometric detail. SDFs have since been instrumental in real-time rendering, with Quilez Quilez (2008) popularizing their use in procedural ray marching and Green Green (2007) introducing their application for scalable and anti-aliased text rendering. Advances in GPU optimization, as explored by McGuire et al. McGuire et al. (2013), have further enhanced the real-time capabilities of SDFs. Techniques like the Marching Cubes algorithm Lorensen & Cline (1987) and Dual Contouring Ju et al. (2002) enable the conversion of volumetric SDFs into detailed meshes, critical for physics simulations and surface reconstruction. In physics-based applications, Fournier and Rico Fournier & Rico (2006) leveraged SDFs for efficient collision detection and soft body interactions. More recently, neural approaches like DeepSDF Park et al. (2019) and NeRF-SDF Yariv et al. (2020) have demonstrated the power of combining SDFs with deep learning for shape representation and neural rendering. Furthermore, sparse data structures, such as OpenVDB Museth (2013), address memory and computation challenges in large-scale volumetric datasets. These works underscore the importance of SDFs in advancing real-time rendering, procedural modeling, and computational efficiency, forming the foundation upon which this research builds. In our work, we highlight the differentiable property of the SDF to generate math figures using math conditions.

## M    INTERACTIVE ONLINE TOOL

To promote broader use and facilitate further research, we have developed an interactive online tool based on GeoSDF. This tool enables the community to synthesize geometric diagrams by providing input as either natural language text or Python code. It serves as a practical demonstration of our framework and a valuable resource for researchers and educators. A demonstration video showcasing the tool's functionality is included in the supplementary materials.

## N    COMPARE WITH GEOMETRIC CONSTRAINT SOLVING (GCS)

From the task perspective. For the CAD related works, it is interactive geometric correction. The goal is to take a user's rough engineering sketch and "snap" it to mathematical precision while maintaining the user's original intent. The mathematical frameworks of prior methods lack the mechanism to resolve Topological Ambiguity. They rely on the user's sketch to provide the "topology" (the relative ordering and approximate relationships of elements), using the solver only to refine the "metric" properties (exact lengths and angles). More specific, The prior works Bouma et al. (1995) and Ge et al. (1999) explicitly state that their method is "sensitive to initial values" and handles under-constrained problems by finding a solution where the deviation from the initial guess, $||X - X_0||$, is minimized. Owen (1991) relies on decomposing a constraint graph into solvable subgraphs. The author explicitly admits that for under-constrained configurations, his algebraic solver "cannot be guaranteed that a solution will be found even though it exists".

From the methodological perspective, our method provide a better Gradient Quality to achieve automated generation from scratch. The SDF representation allows us to define a smooth differentiable loss landscape. This allows the constraints to converge reliably from random noise. While all prior works do not provide Constraint Functions in the modern sense of differentiable. They provide raw algebraic equalities. In these works, a "constraint" is just a *hard* math equation that must equal zero. They don't have "functions" that guide a generation process. They treats constraints as a system of polynomials $F(X) = 0$. For example, they decompose a "tangency" constraint into primitive algebraic checks like "collinear" or "equal distance". Therefore, an algebraic equation is either "Satisfied" ($= 0$) or "Broken" ($\neq 0$). It doesn't inherently tell you *how* to fix it. In contractive, we provide the *soft* Constraint Functions. These are optimization modules, rather than raw coordinate arithmetic. For example, the constraint "Inside": [1], [2], and [3] are impossible to represent easily. "Inside" is an inequality (topology), not a polynomial equality. We define 'Inside(Point, Polygon)' using SDF logic (negative distance), allowing you to optimize for topological relationships that algebra struggles to define.

In conclusion, our method is a Generative method from scratch, but the CAD literature need a good starting point. Our method advances this by formulating constraints as differentiable semantic functions. Unlike algebraic equalities, our functions (e.g., Inside(), Crowd()) provide continuous gradient landscapes that actively guide the optimization process. This allows us to handle complex topological constraints (like containment or convexity) that are mathematically intractable for the strictly algebraic formulations of prior work.

## O    IMO BATCH VISUALIZATION

We show two IMO batch visualization cases. In the Fig. 16 and Fig. 17, we can see the synthesis results are accurate and geometrically consistent.

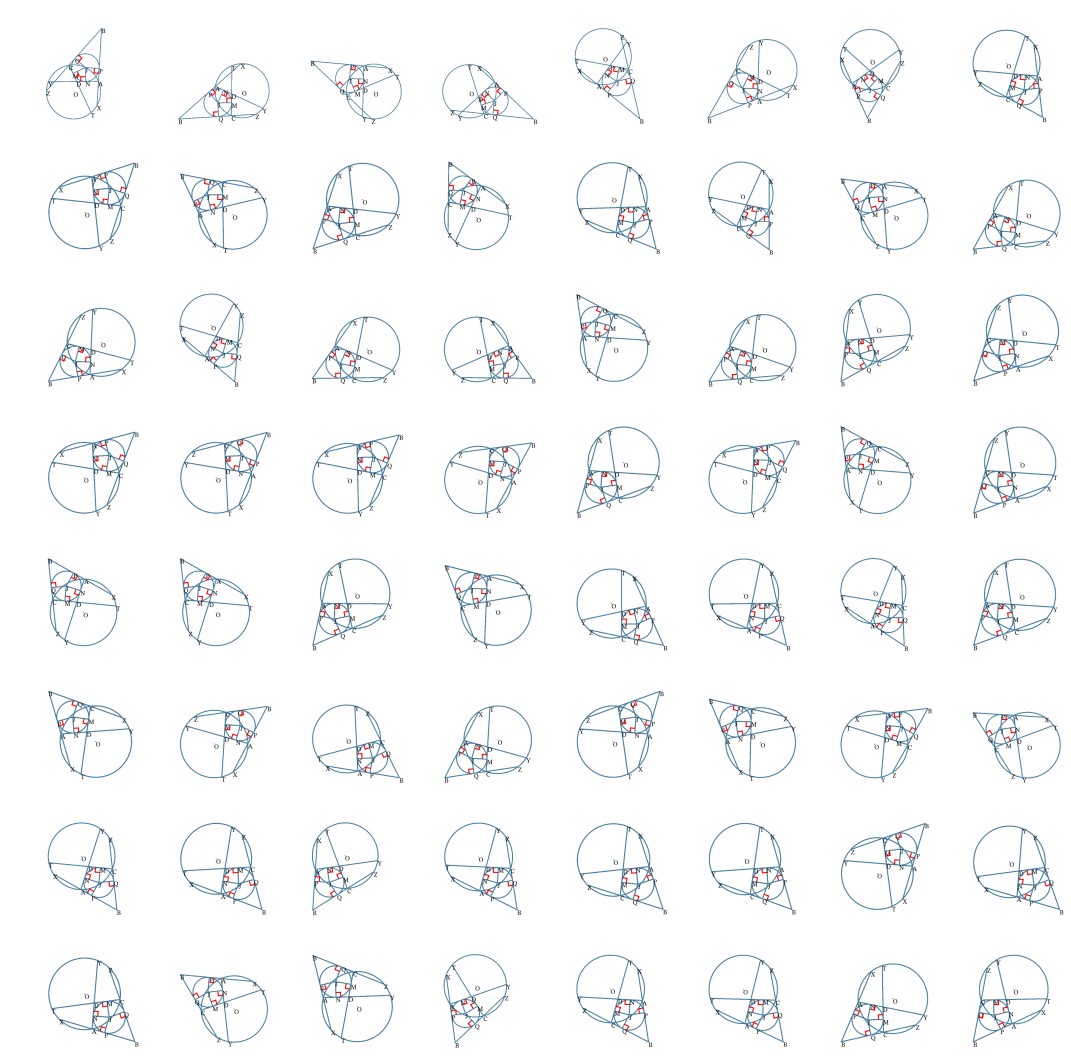

Figure 16: A batch of synthesis diagrams. Source from IMO 2021 Problem 4.

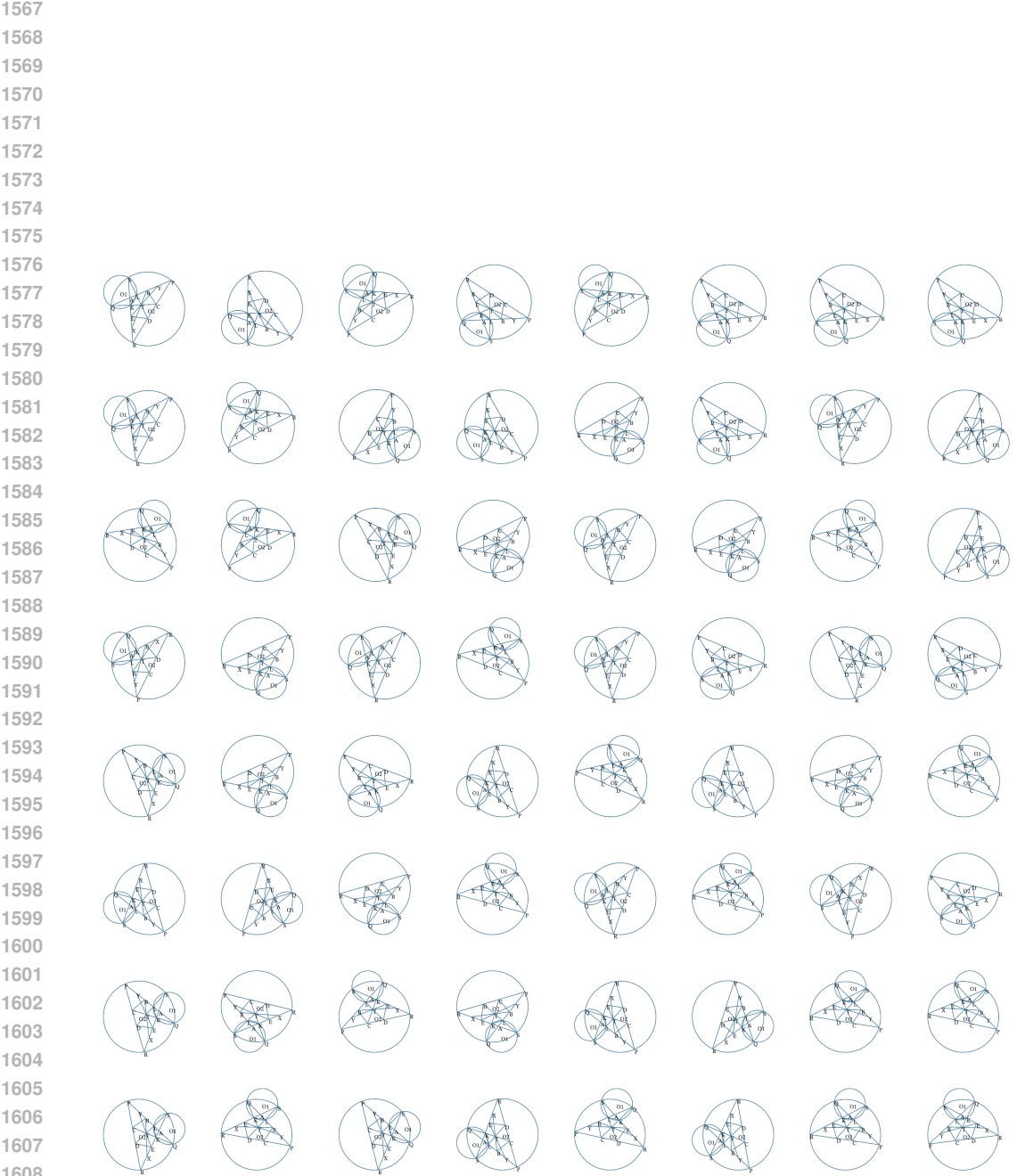

Figure 17: A batch of synthesis diagrams. Synthesis from IMO 2022 Problem 4.

# P    QUALITATIVE COMPARISON

We compare the synthesized diagrams to the ChatGPT output in the IMO problem set. The rule-based method cannot take any user input, so it cannot synthesize the IMO problem diagrams, so we do not consider it in this section. The prompt input to ChatGPT is "Please create a math diagram based on this problem statement". From the figure 18, the ChatGPT results are not accurate for the elements and the constraints.

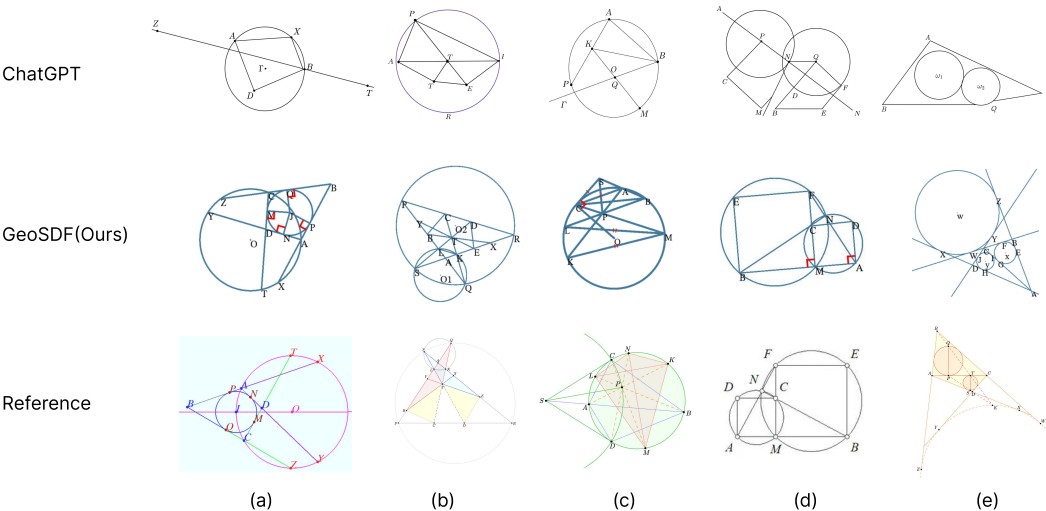

ChatGPT

GeoSDF(Ours)

Reference

(a)          (b)          (c)          (d)          (e)

Figure 18: The five diagrams are compared between GeoSDF (ours) and ChatGPT (22 Nov 2025, latest version). The reference diagrams are for reference only, because they may be incomplete or there may be multiple forms.(a) 2021 p4, (b) 2022 p4, (c) 2010 p4, (d) 1959 p5, (e) 2008 p6.

## Q  QUALITATIVE COMPARISON FOR CROWD REGULARIZATION TERM

The $\tau_r$ and $\lambda_{crowd}$ stems from the theoretical design of the crowd regularization as a vanishing penalty function. Mathematically, the term defined by $\sum [\max(0, \tau_r - ||x_i - x_j||)]^2$ acts solely as a repulsive barrier against degenerate geometric collapses; once the Euclidean distance between elements exceeds the threshold $\tau_r$, the regularization loss and its gradient become identically zero. Consequently, in the final converged state where elements are spatially distinct, the regularization term vanishes from the total objective, rendering the weight $\lambda_{crowd}$ inert regarding the final geometric precision. Furthermore, due to the inherent scale invariance of Euclidean geometry, variations in $\tau_r$ merely encourage the optimizer to converge to a spatially larger instance of the diagram to satisfy the separation constraint, without conflicting with the angular or ratio-based constraints that determine synthesis accuracy.

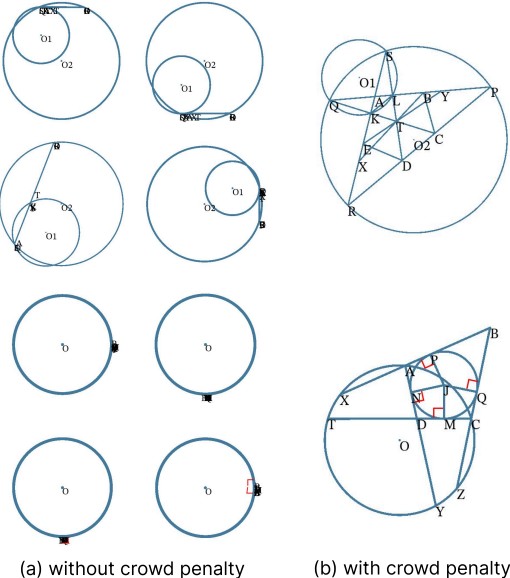

(a) without crowd penalty  (b) with crowd penalty

Figure 19: Qualitative Comparison for Crowd Regularization term in two IMO diagrams. (a) w/o crowd term, (b) w/ crowd term. The elements in the diagram tend to cluster together, resulting in a trivial global minimum solution. The Crowd Regularization effectively pushes all elements away.

## R  SYNTHESIS OF IMPLICIT CONSTRAINTS AND GEOMETRIC DISCOVERY

To address the discussion on implicit assumptions and the connection to classical algebraic methods (e.g., Wu's method), we demonstrate GeoSDF's capability to synthesize "missing" constraints through its generation-by-optimization paradigm. Unlike algebraic methods that derive missing assumptions symbolically, GeoSDF allows implicit geometric properties to emerge naturally as the system converges to a valid Euclidean configuration.

As an example, we illustrate this capability using the construction of a Rhombus. A rhombus is explicitly defined by the constraint that all four sides are of equal length. A fundamental, yet implicit, property of a rhombus is that its diagonals intersect at right angles (perpendicularity). We initialized a geometric optimization with four points A,B,C,D to form a quadrilateral. The loss function included only the following explicit constraints: $AB = BC = CD = DA$ and Point P on $AC$ and $BD$. Critically, we did not include any constraint enforcing the perpendicularity of diagonals AC and BD. We ran the optimization using 8 distinct random seeds to ensure robustness.

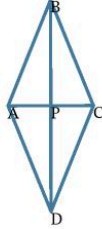

Figure 20: The reference diagram.

Upon convergence, we extracted the geometric properties of the synthesized diagrams. We measured the angle formed by the intersection of diagonals AC and BD. The measured angle consistently converged to $\theta_{diagonals} = 1.570642$ in radians, in this figure 20. Comparing this to the mathematical constant for a right angle, the error is negligible ($\Delta \approx 1.510^{-4}$). Also, we take the IMO 2022 P4 and 2021 P4 as the experiments, which still conform.

Although the perpendicularity constraint was "missing" from the input description, GeoSDF successfully synthesized a diagram where this property holds true. This demonstrates that by optimizing for a subset of necessary premises, the framework naturally settles into a configuration where implicit assumptions are satisfied. This "verification-by-construction" property allows GeoSDF to identify and verify geometric truths that were not explicitly programmed, offering a numerical parallel to the symbolic synthesis found in Wu's method.

# S USE OF LARGE LANGUAGE MODELS

In accordance with the submission guidelines, we declare the usage of large language models (LLMs) or other AI-powered generative tools in this paper. We employed ChatGPT 5 to help polish the language and improve the readability. No LLMs were involved in designing experiments, analyzing data, or contributing to the scientific findings of this work.

In the experiment, we use the Qwen2.5-7B-Instruct to build a natural language parser. Here are the details: (a)Model: We utilized Qwen2.5-7B-Instruct. (b) Compute: Fine-tuning was performed on 4 x A800 80G GPUs for 3 epochs. (c) Data: We used the text-to-constraint pairs from FormalGeo7k.

