# OpenReview forum: "GeoSDF: Plane Geometry Diagram Synthesis via Signed Distance Field"
_ICLR.cc/2026/Conference — Submitted to ICLR 2026_

### Official Review · Reviewer_s9eZ · 2025-10-25

**Soundness:** 4
**Presentation:** 3
**Contribution:** 4
**Rating:** 8
**Confidence:** 3

**Summary:**

This paper introduces GeoSDF, a framework for synthesizing plane geometry diagrams directly from natural-language problem statements. The method converts text into symbolic geometric constraints using an LLM, represents geometric primitives as Signed Distance Fields, and jointly optimizes them to satisfy all constraints. The system produces mathematically valid, self-verifiable diagrams and can even solve geometry problems via direct measurement of optimized configurations. Experiments on FormalGeo7k, GeoQA, and IMO problems demonstrate significant gains over prior rule-based and model-based approaches.

**Strengths:**

One of the most significant contributions of this paper is the adoption of Signed Distance Fields (SDFs) as the foundational representation for plane geometry diagrams. This choice is both clever and original: by expressing points, lines, and circles as differentiable SDFs and formulating geometric constraints as differentiable loss terms, the method transforms symbolic geometric reasoning into a continuous optimization problem. This formulation elegantly unifies geometry synthesis and constraint satisfaction under a single differentiable framework, enabling gradient-based optimization and smooth convergence toward valid configurations. The approach bridges ideas from computer graphics (implicit surfaces) and symbolic reasoning, marking a clear conceptual advance in geometric problem solving.

Besides of differentiability, the SDF formulation inherently supports quantifiability—every geometric element (angle, distance, area) can be measured directly from the optimized field. This allows GeoSDF not only to synthesize diagrams but also to verify their correctness and even solve geometry problems by extracting numerical answers (“solve-by-construction”). This self-verification property is a strong differentiator compared to black-box neural or diffusion-based methods, and it provides a principled link between diagram generation and mathematical reasoning.

Moreover, the integration of a fine-tuned LLM (Qwen2.5-7B-Instruct) for natural-language parsing is a practical and impactful design decision. It enables the system to directly interpret geometry problems written in natural language, translating them into symbolic constraint sets without requiring a domain-specific language or manual encoding. The resulting pipeline—from text parsing to symbolic representation to differentiable optimization—is compact, self-contained, and user-friendly. This end-to-end design represents a major step toward fully automated geometry reasoning, offering accessibility for future applications.

Finally, The method proposes a batch-enabled high efficient optimization tools, and achieves state-of-the-art results on GeoQA (95.9% accuracy, +20% over prior SOTA) and performs well on IMO-level problems (88.7% human-verified accuracy).

**Weaknesses:**

Overall the paper is great. There are still several places to improve the final quality:

- While the qualitative results in Figures 3–5 effectively show GeoSDF’s synthesized diagrams, the paper would benefit from more side-by-side visual comparisons against competing systems (e.g., rule-based and diffusion-based baselines). Currently, most comparisons are numerical or textual. Presenting visual outputs from alternative methods under the same problem settings would make the superiority of GeoSDF’s precision and stability more intuitive and convincing.

- Sections 4.6 and 4.7 already provide excellent empirical analyses of convergence and regularization. However, a brief theoretical explanation of why the non-convex optimization behaves robustly, or how the regularization weight $\lambda$ balances constraint accuracy and geometric spacing, would make the methodology more principled and broadly reproducible.

**Questions:**

1. The paper reports strong parsing performance (F1 = 87.74%, Jaccard = 83.53%) when converting natural language to symbolic constraints. Could the authors elaborate on the types of parsing errors that still occur, especially those involving implicit or unstated geometric relationships? In such cases, does GeoSDF attempt any post-processing, constraint repair, or inference to recover missing elements, or are these examples simply discarded during synthesis?

2. How feasible would it be to extend the SDF-based formulation to 3D or analytic geometry? What challenges might arise in terms of representation or optimization efficiency?

---

> ### Author Response · Authors · 2025-11-26
> **Response to Weakness**
>
> > **Weakness 1:** While GeoSDF's qualitative results are effective, the paper needs side-by-side visual comparisons against competing systems to show its superior.
>
> **Response**: Thanks for providing the suggestion to refine visual comparisons against competing systems. We have **added a new comparison figure** at "Appendix P Qualitative Comparison" in the revision paper.
>
>
> > **Weakness 2:** Sections 4.6 and 4.7 already provide excellent empirical analyses of convergence and regularization. However, (1) a brief theoretical explanation of why the non-convex optimization behaves robustly, or (2) how the regularization weight balances constraint accuracy and geometric spacing.
>
> **Response**:
>
> (1) **Theoretical Basis for Robustness**: The robustness of our non-convex optimization stems from the intrinsic properties of SDF. Unlike discrete geometric representations, SDFs define a **continuous and differentiable** scalar field over the domain. The geometric constraints (e.g., distance, incidence) are formulated as smooth potential energy functions over these fields. While the global landscape is non-convex, the local minima corresponding to valid geometric configurations tend to have wide basins of attraction. By employing **batched stochastic initialization** (batched parallel optimization, e.g., batch size = 1024), we effectively sample the landscape, significantly maximizing the probability that at least one seed falls into the convex basin of a valid solution. Finally, we select the best solution from the batch based on constraint satisfaction and visual quality, for which the loss is lower than 0.1.
>
> (2) **Balancing Regularization**: The total loss can be viewed as a physical system **seeking equilibrium** between 'geometric tension' ($L_{constraints}$) and 'crowd repulsion' ($L_{crowd}$). $L_{constraints}$ acts as a hard constraint (high stiffness), driving the system toward the zero-level set of the solution manifold. $L_{crowd}$ acts as a soft regularization (low stiffness), providing a gradient signal only when elements are unnecessarily clustered or degenerate. The weight $\lambda_{crowd}$ balances these forces. which is not sensitive within a reasonable range. We **added experiments** in "4.6.3 Impact of Crowd Regularization" to show the sensitivity analysis of different weights of crowd regularization.

---

> ### Author Response · Authors · 2025-11-26
> **Response to Questions**
>
> > **Question 1:** The paper reports strong parsing performance (F1 = 87.74%, Jaccard = 83.53%) when converting natural language to symbolic constraints. (1) Could the authors elaborate on the types of parsing errors that still occur, especially those involving implicit or unstated geometric relationships? (2) In such cases, does GeoSDF attempt any post-processing, constraint repair, or inference to recover missing elements, or are these examples simply discarded during synthesis?
>
> **Response**:
>
> (1). **Parsing Error Analysis**: As detailed in Appendix D, parsing errors primarily stem from implicit geometric information. For instance, a problem might describe 'Two lines intersecting at A' without explicitly defining the lines first, or imply collinearity without stating it. The LLM parser occasionally fails to infer these unstated prerequisite elements, leading to an under-defined constraint set.
>
> (2). **Handling Strategy**: Currently, GeoSDF does not employ heuristic post-processing to 'repair' constraints, which are thought of as the missing or wrong caused by user input. Instead, we leverage the quantifiability of our SDF framework as a rigorous verification filter. If a parse is semantically incorrect or missing constraints, the optimization process generally fails to converge to a near-zero loss (or converges to a degenerate state detected by the crowd metric). We treat instances with final Loss >0.1 as synthesis failures and discard them. This 'detect-and-discard' strategy ensures that we do not output mathematically invalid diagrams; outputs are always the correct diagrams. To be more specific:
>
> 1. If the semantically incorrect (e.g., AB parallel to CD and AB perpendicular to CD, which are the conflict constraints), there will be no output (all the samples in a batch fail to converge);
> 2. If the missing constraints (e.g., a triangle ABC, the user wants AB perpendicular to BC, but the user misses it), there will be all correct triangles ABC, and the missing constraint is not in the consideration (because not the input to the pipeline). In the experiment, we find that most of the failure cases are caused by the missing constraints (the constraints are not input in the dataset).
>
> Future work will explore using the high residual loss as a feedback signal to prompt the LLM to refine its generated constraints (an iterative 'Self-Correction' loop).
>
> > **Question 2:** (1) How feasible would it be to extend the SDF-based formulation to 3D or analytic geometry? (2) What challenges might arise in terms of representation or optimization efficiency?
>
> **Response**: We appreciate the reviewer's interest in the extensibility of our framework. Extending GeoSDF to 3D and analytic geometry is not only feasible but mathematically natural, as Signed Distance Fields are dimension-agnostic and can represent any implicit surface.
>
> (1). Analytic Geometry: We have **expanded to handle analytic curves**. The only requirement is defining the differentiable distance function for the specific primitive. We have added "Section 4.7" and "Appendix B (Complex Geometric Elements)" in the revised manuscript, which demonstrates the successful synthesis of Hyperbolas and Bézier Curves.
>
> (2). 3D Geometry: We have added a new section, "Appendix J (3D GEOMETRIC DIAGRAMS)", to **demonstrate 3D synthesis**.
> - Representation & Optimization: The representation remains efficient; we simply expand the coordinate space to $\mathbb{R}^3$. Optimization scales linearly with the number of elements, though the search space is larger.
> - Rendering Challenges: The primary challenge in 3D is not synthesis, but visualization. Standard iso-surface extraction methods like Marching Cubes produce opaque meshes, which obscure internal geometric structures (e.g., the center of a sphere). To address this, we bypass iso-surface extraction for the final output; instead, we extract the optimized geometric parameters directly and utilize Plotly to render wireframe-based, transparent visualizations that preserve geometric clarity.

---

### Official Review · Reviewer_i6Ji · 2025-10-27

**Soundness:** 3
**Presentation:** 3
**Contribution:** 3
**Rating:** 6
**Confidence:** 3

**Summary:**

This paper introduces GeoSDF, a framework for automatic plane geometry diagram synthesis using Signed Distance Fields (SDFs). The method parses natural language problem descriptions into a symbolic constraint representation, represents geometric primitives (points, lines, circles) as SDFs, and optimizes them to satisfy geometric relationships via differentiable loss functions. The framework supports self-verification and quantifiable measurement of geometric entities, allowing both diagram generation and problem-solving. Extensive experiments show improvements over existing model- and rule-based systems.

**Strengths:**

The strengths lie in its novel and unified design: it introduces an SDF-based formulation that encodes geometric constraints in a fully differentiable manner, allowing optimization and verification within the same framework.
Empirically, GeoSDF achieves strong results across several benchmarks, clearly outperforming prior neural and multimodal LLM-based solvers.
The outputs are both quantifiable and interpretable, supporting precise visualization and direct geometric reasoning from a shared representation.
Overall, GeoSDF offers a broadly impactful contribution that bridges symbolic reasoning, geometry education, and visual problem-solving in AI-assisted mathematical systems.

**Weaknesses:**

The weaknesses mainly center on generality and robustness. While GeoSDF performs impressively on polygonal and circular figures, it lacks discussion or demonstration of more complex composite geometries—such as spline curves, conic sections, or freeform loci—which limits its expressive scope.
Furthermore, the optimization procedure may struggle with severely underdetermined or overconstrained systems, yet the paper includes few examples analyzing these failure modes. The reliance on a fine-tuned LLM parser also raises concerns about brittleness when faced with ambiguous or partially specified problem statements.
Finally, there is no explicit ablation or stress test on scalability with respect to the number of geometric elements or constraints—a key factor for evaluating practical usability and extensibility.

**Questions:**

I appreciate clarification on several technical aspects of the framework.
In particular, it remains unclear how GeoSDF could be extended to handle curved or parametric primitives, such as splines or general conic sections, and whether the current SDF formulation could accommodate these without major redesign.

Another point of concern is how the optimization behaves in highly complex or deeply nested constraint graphs with numerous interdependent elements, and whether there exists a practical failure threshold beyond which convergence becomes unreliable.

Additionally, clarification is needed on whether the introduced crowd regularization might inadvertently hinder synthesis accuracy in dense configurations where geometric elements are naturally close.

Finally, it would be valuable to discuss the feasibility of extending GeoSDF beyond planar settings to handle non-planar or 3D geometric diagrams, potentially paving the way toward applications in solid geometry.

---

> ### Author Response · Authors · 2025-11-26
> **Response to Weaknesses**
>
> We appreciate the reviewer’s constructive feedback and have addressed each concern below.
>
> > **Weakness 1 & Questions 1:** (W1) Lacks discussion of more complex composite geometries, such as spline curves, conic sections. (Q1) How could GeoSDF be extended to handle the above geometries without major redesign?
>
> **Response**:
>
> (1) The SDF framework is **naturally extensible**. Unlike rule-based constructive geometry, adding a Conic Section (e.g., a Hyperbola) and spline curves only requires defining its differentiable distance function. We have added a preliminary example of a batch synthesis in the "Appendix B Complex Geometric Elements" to demonstrate this flexibility.
>
> (2) Furthermore, we have included a new "Section 4.7" and "Appendix J 3D Geometric Diagrams" in the revised paper to show the **synthesis of 3D geometric diagrams** in the current GeoSDF framework. Our approach can be easily extended to other geometric primitives as long as their SDFs are differentiable.
>
> > **Weakness 2:** Furthermore, the optimization procedure may struggle with severely (1) underdetermined or (2) overconstrained systems, yet the paper includes few examples analyzing these failure modes. (3) The reliance on a fine-tuned LLM parser faced with ambiguous or partially specified problem statements.
>
> **Response**:
>
> (1). **Underdetermined Systems**: GeoSDF handles underdetermined systems (where multiple valid diagrams exist) naturally. The optimization landscape contains multiple zero-energy minima corresponding to different valid configurations. Our batched stochastic initialization allows exploration of these multiple solutions. We select the best solution based on constraint satisfaction and visual quality. In practice, we observe diverse valid outputs in such cases, demonstrating the method's flexibility. To be more specific, if the user inputs a triangle ABC, and wants AB perpendicular to BC, but the user misses it, it will give all correct triangles ABC, and the missing constraint is not in the consideration (because not the input to the pipeline).
>
> (2). **Overconstrained Systems**: In cases where the LLM generates conflicting constraints (making the system unsolvable), the optimization landscape has no zero-energy solution. The solver converges to a local minimum with a non-zero residual loss. We detect this via our convergence threshold (>0.1) and flag the result as invalid, effectively identifying the parser's error. For instance, if user input AB parallel to CD and AB perpendicular to CD, which are the conflict constraints, there will be no output (all the samples in a batch fail to converge).
>
> (3). **Parser Brittleness**: We acknowledge that the end-to-end performance is bounded by the LLM's parsing capability. However, our experimental results in Table 2 show that even with this bottleneck, GeoSDF achieves **SOTA performance** (78.5%). Furthermore, the separation of parsing (LLM) and solving (SDF) allows our framework to immediately benefit from stronger future LLMs without modifying our pipeline.
>
> > **Weakness 3:** Finally, there is no explicit ablation or stress test on scalability with respect to the number of geometric elements or constraints.
>
> **Response**: We have the experiments on the complexity of elements and constraints. Our evaluation on the IMO dataset serves as a stress test for scalability. As noted **in Section 4.6**, we successfully synthesized diagrams for IMO problems containing up to 24 distinct geometric elements and 24 interconnected constraints. Even at this high complexity, the optimization converges in approximately 21.3 seconds on a single GPU (batch size=1024), which is fully vectorized in PyTorch. The **computational cost grows linearly** with the number of constraints $O(∣C∣)$ and elements $O(|E|)$. The parallel batch processing ensures that increasing element count does not disproportionately impact stability, as we maintain a constant probability of successful initialization across the batch. In contrast to the baseline methods (rule-based and model-based), which struggle with the easy problems, GeoSDF consistently synthesizes accurate diagrams in the most difficult challenge.

---

> ### Author Response · Authors · 2025-11-26
> **Response to Questions**
>
> > **Question 2:** Another point of concern is how the optimization behaves in highly complex or deeply nested constraint graphs with numerous interdependent elements, and whether there exists a practical failure threshold beyond which convergence becomes unreliable.
>
> **Response**:
>
> 1. Global Optimization: A key advantage of GeoSDF is that it solves constraints globally rather than sequentially. In deeply nested constraint graphs (where element C depends on B, which depends on A), sequential constructive solvers suffer from accumulating numerical errors. GeoSDF optimizes all variables simultaneously, distributing the error evenly across the graph to find a globally consistent state.
> 2. Reliability and Thresholds: Empirically, we observe that convergence reliability correlates with the ratio of constraints to degrees of freedom. For extremely highly constrained problems (like IMO geometry), the landscape becomes more rugged with more local minima.
> 3. Mitigation via Batching: We address this 'ruggedness' by scaling the batch size. As shown in Table 4 (IMO experiments), while an individual optimization seed might have a lower success rate on complex graphs, the batch success rate remains high (~60% for batch size 64). This indicates that there is no hard 'failure threshold' for the complexity encountered in standard high-school and Olympiad geometry; rather, higher complexity simply necessitates a larger initialization batch to ensure a valid solution is found.
>
> > **Question 3:** Additionally, clarification is needed on whether the introduced crowd regularization might inadvertently hinder synthesis accuracy in dense configurations where geometric elements are naturally close.
>
> **Response**: This is a crucial point regarding the trade-off between geometric precision and visual layout. The crowd regularization (Eq. 2) employs a hinge loss formulation, active only when the distance between elements falls below the threshold $\tau_{crowd}$. Inspired by your comment, we added the analysis in the "Appendix Qualitative Comparison for Crowd Regularization term."
>
> 1. Accuracy is Preserved: In dense configurations where elements are *required* to be close (e.g., complex tangency cases), the geometric constraint loss (Eq. 1) dominates the optimization process. Since the geometric constraints rely on strict equality (or near-equality), they generate stronger gradients than the soft penalty of the crowd regularization. Thus, the elements are not 'pushed away' from their mathematically correct positions; rather, the system finds a global configuration that satisfies the geometry while minimizing overlap.
> 2. Global Scaling Effect: To satisfy both the dense geometric constraints and the crowd regularization simultaneously, the optimizer tends to globally scale up the diagram. By expanding the distances between *all* elements, the relative distance satisfies the threshold $\tau_{crowd}$ without violating the geometric relationships. As the diagram coordinates expand, the fixed pixel-width of the rendered lines appears relatively thinner. This is visualized in Figure 3 (Right, Row 1, Sample 2). This example represents a naturally dense configuration. The optimizer successfully expanded the overall scale to satisfy the crowd term, resulting in high precision (thin lines relative to the shape) without compromising the dense geometric cluster.
>
> > **Question 4:** Finally, it would be valuable to discuss the feasibility of extending GeoSDF beyond planar settings to handle non-planar or 3D geometric diagrams, potentially paving the way toward applications in solid geometry.
>
> **Response**: Thank you for your question about 3D. The 3D geometric diagrams could be synthesized. We add an experiment in the appendix to show a 3D example. Please refer to Appendix J 3D Geometric Diagrams in the revised paper.

---

### Official Review · Reviewer_YgAs · 2025-10-29

**Soundness:** 3
**Presentation:** 3
**Contribution:** 2
**Rating:** 2
**Confidence:** 4

**Summary:**

Given a geometry problem in Natural Language, the problem here is to construct
a 2d planar diagram illustrating the question.  General-purpose models produce
geometrically inconsistent figures.  Rule-based systems too struggle. The approach
presented in this paper does so by standard gradient descent to find the figure
that minimizes some cost. The cost is set up so that minimizing the cost forces
satisfaction of the constraints in the problem description.

The approach is fairly straight forward: the NL description is parsed to
symbolic constraints, which are then turned into an optimization problem
which is solved by gradient descent.

The paper shows that GeoSDF can produce high-quality geometry figures corresponding
to NL problems.

The paper also contains a "solve-by-construction" approach, where the diagram
constructed by the GeoSDF is used to measure the entity being sought in the problem
and thus, directly answer the question. This method is shown to perform very well.

**Strengths:**

Strengths:
1. GeoSDF is a very reasonable approach for constructing geometry diagrams for problems.
2. The evaluation shows that the method works.

**Weaknesses:**

Weaknesses:
1. This is a fairly natural way to create diagrams. Turning symbolic constraints
solving into an optimization problem is also a common technique.
2. The connection to ML is basically through the use of gradient descent for
solving an optimization problem. And then there is the "solve-by-construction" paradigm
discussed next.
3. The construction-based approach, as described here, can only solve certain problems
where one is asked to measure an angle or segment. Furthermore, it is arguably not
"in the spirit". For example, if two angles of a triangle are given and the problem
asks for the third angle; of course, one can draw and measure the third angle, but that
is usually not the intended approach.

Appendix D line 940 contains a discussion on implicit assumptions, which play a crucial
role in geometry problems. In the early days, algebraic methods were successful in
geometry theorem proving (Wu's method) precisely because they had the ability to
synthesize the missing assumptions. It will be interesting to see if GeoSDF can be adapted
to synthesize missing constraints.

**Questions:**

I do not have any questions.

---

> ### Author Response · Authors · 2025-11-26
> **Response to Weakness 1&2**
>
> **Before the response**, **we clarify that GeoSDF is primarily a diagram synthesis framework, with problem-solving as a secondary attribute**. It pioneers LLM-based natural language-to-diagram translation. Furthermore, our continuous SDF constraints ensure superior optimization stability compared to prior discontinuous methods. In addition, we respectfully believe that the concerns **stems from a partial misunderstanding of the scope and technical focus of GeoSDF**, which we clarify below.
>
> > **Weakness 1:** (1) This is a fairly natural way to create diagrams. (2) Turning symbolic constraints solving into an optimization problem is also a common technique.
>
> **Response:** We thank the reviewer for this observation. However, we respectfully disagree with the reviewer’s assessment.
>
> (1) We believe our framework is **fundamentally different from others** in geometry-diagram synthesis. Previous works rely on manual plotting tools (e.g., Matplotlib) or a rule-based combination. In contrast, our approach generates diagrams directly from natural language problem statements and optimizes the geometric constraints to the final diagrams.
>
> (2) While optimization is indeed a common technique, however, to the best of our knowledge, this is **the first work that expresses geometric elements and constraints to an SDF-based optimization problem**. It is noted that SDF is a continuous formulation that provides greater stability during generation, since the optimization trajectory can be smoothly monitored. By leveraging the continuous nature of SDF, our method can ensure that these constraints are accurately enforced throughout the optimization, leading to highly precise and consistent diagram generation.
>
> > **Weakness 2:** The connection to ML is basically through the use of gradient descent for solving an optimization problem.
>
> **Response:** We appreciate the reviewer’s comment. While our method does employ gradient descent, its connection to machine learning goes beyond simply "using gradient descent to solve an optimization problem." **Our contribution lies in framing diagram synthesis within a fully continuous, differentiable representation**, which is something not present in prior geometry systems.
>
> Specifically, prior approaches rely on rule-based or manually engineered operations, constructing diagrams through fixed heuristics rather than optimization. In contrast, our SDF-based formulation **encodes geometric primitives and constraints into a differentiable objective**, enabling gradient-based optimization over the entire diagram.
>
> Moreover, since our work was submitted to the **Applications track**, we believe it appropriately leverages core ML principles, such as differentiability, continuous optimization, and gradient-based inference, even though it does not involve model training in the traditional sense.
>
> The significance of our work lies in its ability to generate large-scale, accurate sets of diagrams using this technique, providing the type of high-quality data that current large multimodal models require.

---

> ### Author Response · Authors · 2025-11-26
> **Response to Weakness 3&4**
>
> > **Weakness 3:** The construction-based approach is arguably not "in the spirit".
>
> **Response:** We appreciate the reviewer’s insight regarding the pedagogical “spirit” of geometry problem solving. We agree that solving geometry problems usually relies on logical deduction, rather than on measuring or constructing diagrams.
>
> However, we would like to clarify that our primary focus is **generating diagrams** directly from problem statements, rather than providing a new problem-solving strategy. The "solve-by-construction" behavior is not intended as the main way to reason about geometry problems. Instead, it arises naturally as a secondary feature of our system and provides a **verification paradigm** that the generated diagrams are accurate and consistent.
>
> > **Weakness 4:** Appendix D, contains a discussion on implicit assumptions, which play a crucial role in geometry problems. In the early days, algebraic methods were successful in geometry theorem proving (Wu's method) precisely because they had the ability to synthesize the missing assumptions. It will be interesting to see if GeoSDF can be adapted to synthesize missing constraints.
>
> **Response:** We thank the reviewer for raising the connection to Wu’s method. Our **GeoSDF is capable of synthesizing "missing" constraints** in a manner similar to Wu's implicit assumptions, as described in our **Generation-by-Optimization and Verification paradigm**. For example, given the premises of a geometric problem, GeoSDF can generate a diagram where the "missing" constraint emerges naturally and can be formally verified. We add a detailed example in the “Appendix R Synthesis of Implicit Constraints and Geometric Discovery”.
>
> However, upon revisiting our manuscript, we realized, and apologize for the potential confusion, that our original intention in Appendix D was to discuss "implicit information", referring to missing entities during LLM parsing (e.g., unstated points or triangles in a problem statement) rather than mathematical implicit assumptions. To avoid confusion for readers, we have revised the manuscript to replace the term "implicit information" with "implicit information during LLM parsing" and provided a clearer explanation of this concept.

---

### Official Review · Reviewer_M6Vj · 2025-11-01

**Soundness:** 3
**Presentation:** 2
**Contribution:** 3
**Rating:** 4
**Confidence:** 4

**Summary:**

This paper proposes a novel framework, GeoSDF, which represents geometric elements (e.g., points, line segments, and circles) using Signed Distance Fields (SDFs) and constructs a set of constraint functions to encode geometric relationships. By optimizing these constraint functions, the framework generates an optimized field of both elements and constraints, from which diagrams can be efficiently and accurately synthesized through rendering. Both qualitative and quantitative experiments demonstrate that GeoSDF can produce high-level geometric diagrams while maintaining very high accuracy in solving geometry problems.

**Strengths:**

1. The paper introduce GeoSDF, a novel and accurate framework for synthesizing plane geometry diagrams by optimizing SDF representation against symbolic mathematical constraints.
2. The experimental results demonstrate the effectiveness of the proposed method. GeoSDF not only synthesizes geometric diagrams that are consistent with the problem statements but also provides quantitatively accurate results, highlighting its strong quantifiability.

**Weaknesses:**

1. The ablation study in Section 4.7 is overly simple. More comprehensive ablation studies as well as qualitative results are suggested;
2. The hyper-parameter \tau_r in Equation 2 is not explicitly specified, and this parameter can affect the optimized SDF results, ultimately influencing the rendered geometric diagrams. It is recommended that the authors provide the value of this hyper-parameter and analyze how its selection impacts the experimental outcomes;
3. The hyper-parameter λ mentioned in Equation 3 is not specified in terms of its value, and in Appendix A, λ also appears in the definition of the parallelism constraint. However, the meanings of λ in these two contexts are different, which may cause confusion. Also, how these hyper-parameters will affect the results is unclear;
4. Other presentation issues:
* In Section 4.2 and the caption of Figure 3, it is stated that the first row on the left represents the original images. However, it should be the first column that corresponds to the original images.
* In Section 4.4, the explanation of Figure 4 is inconsistent with the figure itself. It is unclear whether the angle being calculated is ∠CDE or ∠DFB.
* Section 4 lacks an explanation for subsection 4.3, and the line spacing in Section 4.2 is inconsistent with that of the other subsections.

**Questions:**

1. The paper mentions that points, line segments, and circles are represented using signed distance fields (SDFs), but the specific formulation or implementation details are not provided. Could the authors clarify how exactly each type of geometric element is represented as an SDF? For instance, how is a line segment’s SDF defined, and how is a circle’s SDF formulated with respect to its center and radius?
2. The paper states that the SDF optimization is a non-convex problem and may converge to local minima or degenerate geometric configurations. Could the authors provide statistics on how often such failures occur in practice? Additionally, are there strategies or heuristics employed to improve convergence stability or avoid degenerate results?
3. Given that Section 3.1 fine-tunes Qwen2.5-7B, could you disclose the use of LLMs in the “Use of LLMs/Responsible AI” section and provide the training hyperparameters, data details, and compliance statements?
4. Could you report the error distribution for angle/length measurements and analyze the sensitivity to the “loss < 0.1” threshold? In addition, how does accuracy change when sweeping the threshold over {0.01, 0.03, 0.1}?
5. Could you detail how the 224k synthetic dataset is used across training/validation/test splits, and provide the deduplication scripts as well as leakage audit results for GeoQA/IMO?

---

> ### Author Response · Authors · 2025-11-26
> **Response to Weakness**
>
> We sincerely thank the reviewer for the insightful comments, and below we address the concerns.
>
> > **Weakness 1:** The ablation study in Section 4.7 is overly simple. More comprehensive ablation studies, as well as qualitative results, are suggested.
>
> **Response**: We thank the reviewer for the suggestion. To address this, we have **added additional qualitative results** in "Appendix Q: Qualitative Comparison for Crowd Regularization Term" to complement the ablation studies in Section 4.6 (Section 4.7 in the original paper). These results provide further support and insight into the impact of the Crowd Regularization term.
>
> > **Weakness 2:** Specify $\tau\_r$ in Equation 2 and analyze its impact on the experimental results.
>
> **Response**: We thank the reviewer for pointing out the need to clarify the hyperparameter $\tau_r$. In the revised paper, we now explicitly provide its value and include a sensitivity analysis. Our experiments show that **$\tau\_r$ is insensitive** across a wide range of settings and does not noticeably affect the optimized SDF or the final rendered diagrams. The corresponding results have been added to “Appendix F: Hyperparameters”, and additional qualitative examples are included in Appendix Q.
>
> For your convenience, here is the $\tau_r$ related experiments in the table below. We tested the value from 0.01 to 0.5, and it is not sensitive.
>
> | Hyperparameter |  Value  |   Acc. (GeoQA test)   |
> | :------------: | :-----: | :------: |
> |    $\tau_r$    |  0.01   |   94.6   |
> |                |  0.05   |   94.6   |
> |                | **0.2** | **94.5** |
> |                |   0.5   |   94.5   |
>
> > **Weakness 3:** The value of λ in Equation 3 is unspecified. Additionally, the reuse of λ in Appendix A for the parallelism constraint creates ambiguity. Disambiguate the notation, provide the values, and analyze their sensitivity.
>
> **Response**: We appreciate the reviewer’s observation regarding the overloaded notation. The symbol $\lambda$ in Equation 3 and the $\lambda$ used in the parallelism constraint in Appendix A represent different hyperparameters. To avoid confusion, we have renamed the parallelism coefficient to $\lambda_{\text{parallel}}$.
>
> We also provide the values and sensitivity analyses of all hyperparameters, including $\lambda$ in Eq. 3 in Section 4.6. Our experiments indicate that these hyperparameters exhibit similar **insensitivity** patterns. From the table below, the $\lambda_{crowd}$ is insensitive from 1.0 to 20.0. Additional qualitative comparisons are included in Appendix Q.
>
> |  Hyperparameter   |  Value   |   Acc. (GeoQA test)  |
> | :---------------: | :------: | :------: |
> | $\lambda_{crowd}$ |   1.0    |   94.9   |
> |                   |   5.0    |   94.5   |
> |                   | **10.0** | **94.5** |
> |                   |   20.0   |   94.3   |
>
>
>
> > **Weaknesses 4:** Some presentation issues.
>
> **Response**: Thank you, we have carefully reviewed the entire paper to double-check all presentation-related issues.

---

> ### Author Response · Authors · 2025-11-26
> **Response to Question 1&2**
>
> > **Question 1:** Could the author clarify how exactly each type of geometric element is represented as SDF for points, line segments, and circles?
>
> **Response**: Thank you for pointing out the question. We understand your concern, we **add the details of the implementation of geometric elements** in "Appendix A" in the revision paper, which includes explicit mathematical formulations for all primitives. For your convenience, we would like to add some of the definitions as follows:
>
> The representation of geometric primitives using SDF, where each element defines a scalar field $f(p)$ representing the Euclidean distance from a query point $p$ to the geometry.
>
> - A **point** with center $c$ is defined by the norm $f_{point}(p;c)=||p-c||\_{2}$;
> - A **circle** with radius $r$ is defined by the absolute difference $f_{circle}(p;c,r)=|||p-c||\_{2}-r |$;
> - A **line** to be defined as $f_{line}(p;a,b)=||v_{ap}-h\cdot v_{ab}||\_{2}$, which defined by points $a$ and $b$, utilize a projection scalar $h=\frac{v_{ap}\cdot v\_{ab}}{||v_{ab}||\_{2}^{2}}$;
> - A **segment** to be defined by clamping the projection to $\overline{h}=clamp(h,0,1)$ resulting in $f_{segment}(p;a,b)=||v_{ap}-\overline{h}\cdot v_{ab}||\_{2}$;
> - Finally, complex structures are created via **composition** using the union operation: $f_{union}(p)=min(f_{A}(p),f_{B}(p),...,f_{N}(p))$.
>
>
> > **Question 2:** Given the non-convex nature of SDF optimization, please quantify the frequency of convergence to local minima or degenerate configurations. Additionally, discuss any heuristics employed to enhance stability.
>
> **Response**: We conducted new experiments on batch size to solver accuracy. Please refer to the table below, which has also been added in "Section 4.6.3" in the revision paper. The bold text is our default value. Our failure rate is 5.5% in a batch size of 32.
>
> It is noted that we have already used the strategies that **parallel running on a large number of initializations** to avoid converging to local minima in the GeoSDF. Without such a strategy, the failure rate suffers a significant increase from $5.5$\%to $29.6$\%, as one-shot synthesis may not fully converge. The reasons are as follows: with the batch size $N$, there will be $N$ times for GeoSDF to converge on each synthesis input. Practically, a larger batch size leads to a lower failure probability of each problem.
>
> | Hyperparameter | Value  |   Acc.  (GeoQA test) |
> | :--: | :-: | :--: |
> |  Batch Size  |   1    |   70.4   |
> | |   2    |   81.9   |
> |  |   4    |   89.8   |
> |  |   8    |   92.7   |
> |  |   16   |   93.7   |
> | | **32** | **94.5** |
> |  |   64   |   94.5   |

---

> ### Author Response · Authors · 2025-11-26
> **Response to Question 3&4&5**
>
> > **Question 3:** Could you disclose the use of LLMs?
>
> **Response**: Thank you for the detailed comments. We have added the following details to the "Use of LLMs" section in the revised manuscript "Appendix S USE OF LARGE LANGUAGE MODELS":
>
> - Model: Qwen2.5-7B-Instruct.
> - Hardware: Fine-tuning was performed on 4 x A800 80G GPUs for 3 epochs using LoRA.
> - Data: We used the text-to-constraint pairs from FormalGeo7k.
>
> > **Question 4:** Could you report the error distribution for angle/length measurements and analyze the sensitivity to the “loss < 0.1” threshold? In addition, how does accuracy change when sweeping the threshold over {0.01, 0.03, 0.1}?
>
> **Response**: We thank the reviewer for this rigorous inquiry. Follow your request, we have added the error distribution and the sensitivity analysis about the loss threshold $\tau\_c$. The detailed analysis has been added in the "Section 4.6.3". For your convenience, we show the table below. From the table, the **$\tau\_{c}$ is stable from 0.01 to 1**.
>
> **Error distribution**: We statistic the error distribution as follows: Angle: 1.9%, Distance: 0.65%, Area: 2.4%, Others: 0.49%, and Total: 5.5%.
>
> | Hyperparameter |  Value  |   Acc.   |
> | -| - |-  |
> |   $\tau_{c}$   |  0.01   |   92.5   |
> | |  0.03   |   94.1   |
> | | **0.1** | **94.5** |
> | |    1    |   94.8   |
>
>
>
> > **Question 5:** Could you detail how the 224k synthetic dataset is used across training/validation/test splits, and provide the deduplication scripts as well as leakage audit results for GeoQA/IMO?
>
> We would like to clarify that **our dataset is an extension of the original FormalGeo7k dataset**. In FormalGeo7k, each problem consists of a single problem statement paired with one diagram. In our 224k synthetic dataset, for each problem statement, we provide 32 distinct diagrams.
>
> Since the original FormalGeo7k dataset does not provide explicit training, validation, or test splits, we similarly do not define such splits in our extension. While we do not provide additional tools to fully prevent duplicates, the original FormalGeo7k dataset has been extensively used and evaluated by the research community, and we trust that it maintains high quality.
>
> Fortunately, we have also designed several measures to minimize potential data leakage. For each diagram, we annotate the corresponding source ID from its original dataset. Users can utilize these source IDs to manually control or filter potential duplicates, ensuring that specific training, validation, or test splits remain free of leakage problems during their experiments.

---

### Comment · Area_Chair_MXTC · 2025-11-23
**Geometric Constraint Solving**

The authors appear to be unaware with the research field of Geometric Constraint Solving, which focuses on generating geometric diagrams from their constraints; see, for example:

Algebraic solution for geometry from dimensional constraints
JC Owen - Proceedings of the first ACM symposium on Solid …, 1991 - dl.acm.org

Geometric constraint solver
W Bouma, I Fudos, C Hoffmann, J Cai, R Paige - Computer-aided design, 1995 - Elsevier

There exist methods of Geometric Constraint Solving based on optimization; see, for example:

Geometric constraint satisfaction using optimization methods
JX Ge, SC Chou, XS Gao - Computer-Aided Design, 1999 - Elsevier

These works should be compared.

---

> ### Author Response · Authors · 2025-11-26
> **Response to AC**
>
> We thank the AC for bringing Geometric Constraint Solving (GCS) to our attention. Although the GCS literature is not typically considered part of the related work in our community [1-3] all these years, it is indeed relevant, and we carefully reviewed the papers suggested by the AC. Upon examination, the fundamental difference between the two frameworks lies in their mathematical representation, reliance on initialization, and application scope.
>
> 1. While Ge et al. employ an algebraic equation system that is **highly sensitive to initial values** and **depends on a user-provided sketch** to refine metric properties, GeoSDF leverages SDF to **generate diagrams autonomously from random noise** via differentiable loss functions.
> 2. GeoSDF **advances the methodological approach used in** GCS, as it treats constraints as soft optimization modules rather than hard polynomial equalities, enabling the resolution of complex topological relationships (e.g., containment, convex) that are intractable for Ge et al.’s algebraic approach.
> 3. Additionally, our method takes natural language problem statements as input, and we could **easily apply to the 3D geometric**.
>
> In short, we thank the AC again for pointing out GCS to the community. In our revision, we cite and discuss these relevant works in “Appendix N: Comparison with Geometric Constraint Solving (GCS)”.
>
>
> Reference:
>
> [1] Renrui Zhang, Xinyu Wei, Dongzhi Jiang, Ziyu Guo, Yichi Zhang, Chengzhuo Tong, Jiaming Liu, Aojun Zhou, Shanghang Zhang, Peng Gao, and Hongsheng Li. MAVIS: Mathematical visual instruction tuning with an automatic data engine. ICLR, 2025. URL https://openreview.net/forum?id= MnJzJ2gvuf.
>
> [2] Deng, L., Zhu, L., Liu, Y., Wang, Y., Xie, Q., Wu, J., … Bai, X. (2025). Theorem-Validated Reverse Chain-of-Thought Problem Generation for Geometric Reasoning. EMNLP, 2025. URL https://aclanthology.org/2025.emnlp-main.38/
>
> [3] Shi, H., Xu, Z., Wang, H., Qin, W., Wang, W., Wang, Y., … Wang, H. (2024). Continual Learning of Large Language Models: A Comprehensive Survey. ACM Computing Surveys, 2025. URL https://dl.acm.org/doi/10.1145/3735633

---

### Author Response · Authors · 2025-11-26
**To AC and all the reviewers**

We appreciate the insightful comments from AC and the reviewers. We believe that all the concerns have been addressed and highlighted the updated parts in the revised paper.

Notably, building on the reviewers' suggestion, we have extended GeoSDF beyond planar settings to handle Non-Planar and 3D Geometric Diagrams, potentially paving the way toward applications in solid geometry, while other frameworks cannot. We include additional discussions and examples in the revised version (see Section 4.7, Appendix B and J).

Thanks for your review again! We hope our responses can address your concerns.

---

### Author Response · Authors · 2025-12-03
**Summary of Rebuttal**

We sincerely thank the Area Chair and all reviewers for their time, insightful comments, and constructive feedback. We appreciate their recognition of GeoPDF’s strengths. Through extensive additional experiments and theoretical analyses, we have addressed concerns regarding **method extensibility (3D and Analytic Geometry)**, **hyperparameter sensitivity**, **visual comparisons**, and **methodological distinctions**.

---

### Strengths from Reviewer

We summarize the key strengths of GeoSDF highlighted by the reviewers below:

- **Novel & Unified Representation (Reviewers s9eZ: 8, i6Ji: 6 , M6Vj: 4, YgAs: 2)**:The representation of Signed Distance Fields (SDFs) is recognized as a clever, reasonable and original formulation that transforms symbolic geometric reasoning into a continuous, differentiable optimization problem .
- **Quantifiability & Self-Verification (Reviewers s9eZ: 8, i6Ji: 6, YgAs:2, M6Vj: 4):** The framework’s ability to directly measure geometric entities enabling a "solve-by-construction" paradigm allows for rigorous self-verification, serving as a strong differentiator from black-box generative models .
- **SOTA Performance (Reviewers s9eZ: 8, i6Ji: 6):** GeoSDF achieves state-of-the-art results on the GeoQA benchmark and demonstrates the capability to synthesize complex IMO-level diagrams where prior rule-based methods fail.

---

### Summary of Revision

We have substantively addressed the specific concerns raised during the review process:

- **Extensions to 3D and Analytic Geometry; Add Visual Comparisons (Reviewers s9eZ: 6; i6Ji: 8):** Both reviewers are interested in expanding the scope and comparisons of our work. In response, we have successfully extended GeoSDF to **Analytic Geometry** (e.g., Hyperbolas, Bézier curves) and **3D Geometric Diagrams**, with new experiments detailed in **Section 4.7** and **Appendix J**. Additionally, we added side-by-side visual comparisons against competing systems in **Appendix P** to intuitively demonstrate our superior precision.
- **Clarifying Hyperparameters (Reviewer M6Vj: 4):** We addressed concerns regarding the specification and sensitivity of hyperparameters (e.g., $\tau_r$, $\lambda_{crowd}$). We added a comprehensive sensitivity analysis in **Section 4.6.3**, demonstrating that the framework remains robust across a wide range of values, and included qualitative results for the Crowd Regularization term in **Appendix Q**.
- **Clarifying Contributions & Methodology (Reviewer YgAs: 2):** We addressed the misconception that our contribution is limited to applying gradient descent. We clarified that our core contribution is the continuous SDF formulation which enables stable generation. Furthermore, regarding the critique on the "spirit" of geometry solving , we clarified that our "solve-by-construction" paradigm primarily serves as a verification mechanism. We also added **Appendix R** to demonstrate GeoSDF's ability to synthesize implicit constraints.
- **Comparison with Geometric Constraint Solving (Area Chair):** We thank the AC for highlighting the Geometric Constraint Solving (GCS) literature. We have added a detailed discussion in **Appendix N**, differentiating our soft, differentiable SDF approach from the rigid algebraic constraints used in traditional GCS. We emphasize that GeoSDF better handles topological ambiguities and initialization sensitivity compared to algebraic solvers.

We believe these revisions have significantly strengthened the paper, demonstrating that GeoSDF is not only accurate and efficient but also highly extensible. We hope our detailed responses and additional experiments have satisfactorily addressed all concerns.



Best regards,

The Authors of GeoSDF

---

### Meta-Review · Area_Chair_SnDE · 2026-01-06

**Summary:**

The core concerns revolve around the perceived novelty and contribution of the method. Reviewers, particularly Reviewer YgAs and the Area Chair, questioned whether the core technical approach, formulating geometric constraints as an SDF-based optimization problem solved with gradient descent, constitutes a significant advance over established techniques in Geometric Constraint Solving (GCS) and other optimization-based methods. While the results are strong, there is skepticism about the fundamental methodological innovation, with some viewing it as a straightforward or natural application of existing concepts. Additional concerns included the scope of the "solve-by-construction" paradigm, hyperparameter sensitivity, and the need for more comprehensive comparisons and ablation studies.

**Reviewer Concerns:**

The rebuttal substantively addressed many specific, technical concerns. The authors provided detailed hyperparameter sensitivity analyses, error distributions, failure rate statistics, and clarified implementation details. They also expanded the paper's scope by demonstrating extensions to 3D geometry and analytic curves, and added visual comparisons. However, the most significant outstanding concern is the perceived lack of fundamental novelty relative to the GCS literature highlighted by the Area Chair. While the rebuttal draws distinctions (soft constraints, differentiability, handling topological ambiguity), it may not fully convince reviewers who view the core idea of optimizing a constraint-based energy function as an incremental application of known principles rather than a paradigm shift. The philosophical concern about the "spirit" of geometry solving, while acknowledged, remains a subjective point of debate.

**Reviewer Scores:**

All reviewers did not response during the rebuttal period.

---

### Decision · Program_Chairs · 2026-01-26

Reject